# Performative Risk Control: Calibrating Models for Reliable Deployment under Performativity

**Victor Li[1], Baiting Chen[2], Yuzhen Mao[3], Qi Lei[1], and Zhun Deng[4]**

[1]New York University
[2]University of California, Los Angeles
[3]Stanford University
[4]University of North Carolina at Chapel Hill

## Abstract

Calibrating blackbox machine learning models to achieve risk control is crucial to ensure reliable decision-making. A rich line of literature has been studying how to calibrate a model so that its predictions satisfy explicit finite-sample statistical guarantees under a *fixed*, *static*, and unknown data-generating distribution. However, prediction-supported decisions may influence the outcome they aim to predict, a phenomenon named *performativity* of predictions, which is commonly seen in social science and economics. In this paper, we introduce *Performative Risk Control*, a framework to calibrate models to achieve risk control under performativity with provable theoretical guarantees. Specifically, we provide an iteratively refined calibration process, where we ensure the predictions are improved and risk-controlled throughout the process. We also study different types of risk measures and choices of tail bounds. Lastly, we demonstrate the effectiveness of our framework by numerical experiments on the task of predicting credit default risk. To the best of our knowledge, this work is the first one to study statistically rigorous risk control under performativity, which will serve as an important safeguard against a wide range of strategic manipulation in decision-making processes.[1]

## 1 Introduction

We have entered an era characterized by the ubiquitous deployment of increasingly complex models, such as large language models (LLMs) with billions of parameters [1–3]. These models play important roles in our daily lives, informing us, shaping our opinions, and deciding allocation of societal resources. However, in most scenarios, these models can only be treated as a blackbox, either because they are too complex to understand or simply because their details are kept private by companies. Given their influence in today's society and their blackbox nature, it is urgent to derive new tools to ensure reliable deployment.

A recent line of work [4–6] has been investigating methods to calibrate the predictions of blackbox machine learning models, generally known as (conformal) risk control. The goal is to ensure reliable deployment by satisfying explicit, finite-sample statistical guarantees (either with high probability or marginally) for controlling risk. The introduced frameworks are lightweight, agnostic to the data-generating distribution, and do not require model refitting. Specifically, in (conformal) risk control, a blackbox model $f$ is given to us, and we need to post-process it using calibration data in order to make our final predictions. The calibration process is governed by a low-dimensional parameter $\lambda$ (e.g., a threshold). For example, [5] and [6] study false negative rate (FNR) control in

---

[1]Code for experiments here: `https://github.com/livctr/rcpp`.

39th Conference on Neural Information Processing Systems (NeurIPS 2025).

tumor segmentation, where $f : \mathcal{X} \mapsto [0, 1]^{d \times d}$ takes an image $x$ as input and outputs scores for all the $d \times d$ pixels in image $x$. For a given $\lambda$, one can produce $\mathcal{T}_\lambda(x) := \{i : f(x)_i \geq 1 - \lambda\}$ to include all the pixels with "high" scores. The aim is to control the expected loss:

$$\mathbb{E}_{(x,y) \sim \mathcal{D}} \left[ 1 - \frac{|y \cap \mathcal{T}_\lambda(x)|}{|y|} \right],$$

where label $y$ is the set of pixels truly containing polyp segmentations, $f(x)_i$ is the $i$-th coordinate of $f(x)$, and $\mathcal{D}$ is the underlying data-generating distribution.

In contrast to tumor segmentation, other applications—particularly those in sociotechnical settings—must account for *performativity*, in which predictions of machine learning models impact the outcome to predict [7]. For instance, a bank may adjust a decision threshold parameter $\lambda$ that governs the bank's loan policies. Such policies then impact the purchasing patterns of the population, which in turn change the features used by the bank in predicting default risk. Similarly, in content moderation, stricter content-filtering policies can affect how users compose their posts. A generation-control parameter in a power plant, which controls unmet demand by setting supply based on demand forecasts, can alter future demand patterns. What unifies these settings is the cyclical relationship between the decision-making parameter $\lambda$ and the distribution it is intended to control risk for. So far, this important setting of risk control on blackbox models under performativity eludes the literature. In light of this, we ***initate*** the study of *Performative Risk Control* (PRC).

**Formalizing our goal.** We post-process the outputs of the pre-trained model $f(x)$ in $\mathcal{Y}$ to generate predictions $\mathcal{T}_\lambda(x) \in \mathcal{Y}'$ indexed by a scalar threshold $\lambda$. Here, $\mathcal{T}_\lambda$ could either be standard predictions, i.e., $\mathcal{Y}' = \mathcal{Y}$, or prediction sets, i.e., $\mathcal{Y}' = 2^{\mathcal{Y}}$. In this paper, we mainly study the expected risk. However, since we further consider the performativity of predictions, the measure of our interest is mainly the *performative expected risk* defined as

$$R(\lambda) := \mathbb{E}_{(x,y) \sim \mathcal{D}(\lambda)} \left[ \ell(y, \mathcal{T}_\lambda(x)) \right],$$

where $\mathcal{D}(\lambda)$ is the distribution induced by the decision threshold $\lambda$ and is typically unknown. In Sec. 4, we also discuss a broader class of measures beyond expected risk. By carefully setting the parameter $\lambda$, we control a user-chosen error rate, regardless of the quality of $f$. Specifically, our goal is to calibrate $f$ on a *calibration* dataset $\mathcal{I}_{\text{cal}}$ (specified later in Sec. 3.1) and achieve the following:

**Definition 1.1.** Let $\hat{\lambda} \in \Lambda$ be the threshold obtained by calibrating on $\mathcal{I}_{\text{cal}}$. We say that risk $R(\hat{\lambda})$ is $(\alpha, \delta)$-performative-risk-controlled if

$$\mathbb{P}(R(\hat{\lambda}) \leq \alpha) \geq 1 - \delta,$$

where the randomness is taken on $\mathcal{I}_{\text{cal}}$. Here, $\alpha \geq 0$ and $\delta \in (0, 1)$ are both specified by the user.

**Our contribution.** Our central contribution is to introduce an iteratively refined procedure to select risk-controlling decision thresholds in a performative environment, as illustrated informally in Fig. 1. Our framework greatly generalizes the previous line of work on risk control for a static distribution to a dynamic and model-dependent distribution. In particular, our work can handle strategic manipulation of input distributions in decision-processes. Our framework satisfies explicit finite-sample statistical guarantees, and we present experimental results that highlight its practical utility. To the best of our knowledge, we are the first to study risk control under performativity, and we believe our work will serve as an important safeguard in a wide range of applications in social science and economics.

## 1.1 Related Work

**Performative prediction.** Performative prediction generalizes the concept of strategic classification [8]—the idea that individuals adjust their features to game a classifier—by formalizing the prediction-influenced data $\mathcal{D}(\theta)$ as a function of the model parameters $\theta$ [7]. Follow-up works on performative prediction have extended the framework to various settings. Miller et al. [9] provides conditions for which performatively optimal model parameters can be found, while Brown et al. [10] considers the history of model deployments when modeling performative distribution shifts. Jagadeesan et al. [11] and Chen et al. [12] cast performativity into a bandit problem, with the assumption of additional knowledge through access to the samples. In these settings, the goal is to find a setting of parameters that in some way minimizes an expected loss and obtain a performative optimal point $\theta_{\text{PO}} = \arg\min_\theta \mathbb{E}_{z \sim \mathcal{D}(\theta)} \ell(z; \theta)$. However, due to the unknown $\mathcal{D}(\theta)$, this task

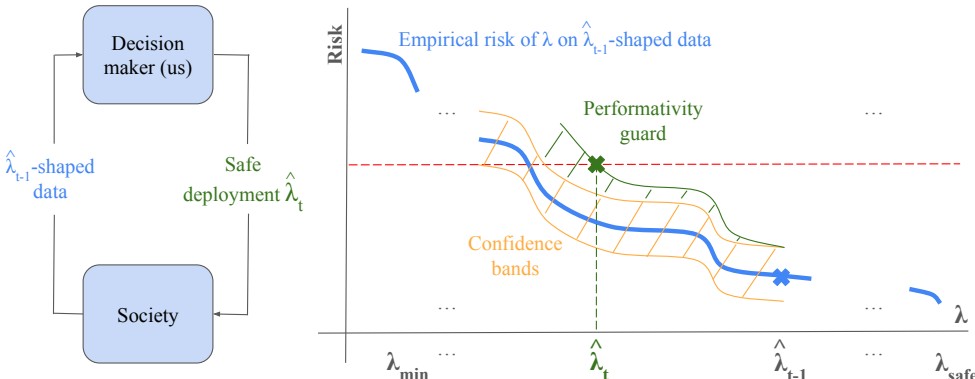

Figure 1: An overview of the Performative Risk Control (PRC) framework. **Left**: We, the decision maker, use samples from past deployments to determine a $\hat{\lambda}_t$ that is safe to deploy in the performative environment. **Right**: We do so by defensively choosing a $\hat{\lambda}_t$, careful to consider generalization error due to sampling and performative error due to deploying in the new distribution induced by $\hat{\lambda}_t$, as further explained in Sec. 3.

is intractable, so much focus has been devoted to studying performative stability, i.e., finding $\theta_{PS} = \arg\min_\theta \mathbb{E}_{z \sim \mathcal{D}(\theta_{PS})} \ell(z; \theta)$. In our setting, our objective is instead to *control* the expected loss, i.e., find the most aggressive setting of the threshold $\hat{\lambda}$ while guaranteeing $R(\hat{\lambda})$ is risk controlled. To the best of our knowledge, risk control in the performative setting has yet to be explored. For an overview on performative prediction, Hardt and Mendler-Dünner [13] serves as a solid reference.

**Distribution-free uncertainty quantification.** Conformal prediction aims to endow black-box models with rigorous, finite-sample statistical guarantees [14–16]. Conformal risk control is an extension to this framework that handles a more general class of loss functions [4, 6]. These works are concerned with controlling the expected risk at some threshold $\alpha$; further work by Snell et al. [17] extends this framework to more general quantile risks.

**Risk control under distribution shift.** The canonical conformal prediction procedure relies on the assumption that the calibration and test data are i.i.d., or more generally, *exchangeable*. Because this assumption often does not hold in real-world environments, statistical extensions of conformal prediction have been developed to incorporate distribution shift. Tibshirani et al. [18] deals with covariate shift, while Gibbs and Candes [19] handles distribution shift in the online setting by continuously re-estimating a single parameter to achieve exact coverage. Barber et al. [20] generalizes conformal prediction to account for nonexchangeability through a procedure that weighs points in the calibration data higher if they are closer in distribution to the test data. Other works focus on making coverage robust to a set of possible test distributions, e.g., those in an $f$-divergence ball around the training distribution [21] and those due to feature alterations by strategic agents after model deployment [22].

## 2 Setup

This section introduces our setting and goals and their significance more formally.

### 2.1 Setting and Notation

For $K \in \mathbb{N}_+$, we use $[K]$ to denote $\{1, 2, \cdots, K\}$. For simplicity, we denote $\mathcal{Z} = \mathcal{X} \times \mathcal{Y}$ and $z = (x, y)$. Recall that we have a prediction function indexed by a threshold $\lambda$, where for a fixed $\lambda \in \Lambda$, $\mathcal{T}_\lambda$ is a function mapping $\mathcal{X}$ to $\mathcal{Y}'$. Here, $\mathcal{Y}'$ could either be $\mathcal{Y}$ or $2^{\mathcal{Y}}$. We further use $\ell(z, \lambda)$ short for $\ell(y, \mathcal{T}_\lambda(x))$.

**Monotonicity.** Throughout the paper, we consider $\ell(z, \lambda)$ to be *non-increasing* with respect to $\lambda$ for any $z \in \mathcal{Z}$. This is mainly following and motivated by previous work in standard risk control [5]. On

one hand, one can think of the case that $\mathcal{T}_\lambda$ is a set-valued function as in [5, 6], that

$$\lambda_1 < \lambda_2 \Rightarrow \mathcal{T}_{\lambda_1}(x) \subseteq \mathcal{T}_{\lambda_2}(x).$$

A larger set includes more candidates and provides a larger tolerance region that will decrease the loss values of loss functions such as the classical one corresponding to tolerance region, i.e., $\ell(y, \mathcal{S}) = \mathbf{1}\{y \notin \mathcal{S}\}$, which satisfies $\mathcal{S} \subset \mathcal{S}' \Rightarrow \ell(y, \mathcal{S}) \geq \ell(y, \mathcal{S}')$. On the other hand, when $\mathcal{T}_\lambda(x)$ is a standard prediction, larger $\lambda$ indicates a more conservative but safer decision. For the example of credit loans, $\mathcal{T}_\lambda(x) = \mathbf{1}\{f(x) \leq 1 - \lambda\}$ may indicate whether an applicant gets credit (1 means they got it). As $\lambda$ decreases, the credit firm relies more on automatic decision-making, accepting a trade-off where it gains efficiency in processing applications but faces increased firm risk as measured by the loss function $\ell(z, \lambda)$.

**Setup for $\Lambda$.** We consider the case where there exists a *safe* threshold $\lambda_{\text{safe}}$ such that $\ell(z, \lambda_{\text{safe}}) = 0$, and we consider $\Lambda = [\lambda_{\min}, \lambda_{\text{safe}}]$ for $\lambda_{\min} \in \mathbb{R}$ [2]. For the example of $\mathcal{T}_\lambda(x) = \mathbf{1}\{f(x) \leq 1 - \lambda\}$ above, we can choose $\lambda_{\text{safe}} = 1$ and in the example of detecting tumor segmentation in [5], we can choose $\lambda_{\text{safe}}$ large enough to make $\mathcal{S}_{\lambda_{\text{safe}}}$ include all possible labels, so that $\ell(y, \mathcal{S}_{\lambda_{\text{safe}}}) = \mathbf{1}\{y \notin \mathcal{S}_{\lambda_{\text{safe}}}\} = 0$.

## 2.2 Our Desiderata and Significance

As mentioned in the introduction, our main goal is to select an *appropriate* threshold $\hat{\lambda}$ by using the calibration dataset $\mathcal{I}_{\text{cal}}$, such that

$$\mathbb{P}(R(\hat{\lambda}) \leq \alpha) \geq 1 - \delta.$$

However, there might be multiple choices for $\hat{\lambda}$ and how to define "appropriate" under performativity is nuanced. We further illustrate our extra desiderata here.

**Achieve user-specified conservativeness.** Our setting prescribes that a larger $\lambda$ leads to being more conservative. As an extreme case, choosing $\lambda = \lambda_{\text{safe}}$ can guarantee zero loss, thus satisfying $(\alpha, \delta)$-performative-risk-controlled for all $\alpha \geq 0$. But this safe-guard choice of $\lambda$ is not useful in practice since it sacrifices too much utility in providing useful class information: for credit scoring, this choice is equivalent to manually reviewing all applications. This illustration demonstrates that a user of this framework may further desire to choose a $\lambda$ that is less conservative. To formalize this, we wish the final choice of $\hat{\lambda}$ also satisfies

$$\mathbb{P}(R(\hat{\lambda}) \geq \alpha - \Delta\alpha) \geq 1 - \delta$$

for a small user-specified $\Delta\alpha$. In particular, we show that our framework allows users to choose $\Delta\alpha$ such that $\Delta\alpha \to 0$ as $n \to \infty$. This means that our framework can provide a $\hat{\lambda}$ also satisfying a tight lower bound guarantee for the performative risk, i.e., $\hat{\lambda}$ is the optimal and least conservative in an *asymptotic* sense.

**Safe at anytime.** Similar to the literature of performative prediction [7], our algorithm is an iterative algorithm (see Sec.3 for details). On a high-level summary, our algorithm guarantees that the $\hat{\lambda}_t$ updated in each iteration is improving, i.e., $\hat{\lambda}_t < \hat{\lambda}_{t-1}$, until we find our final $\hat{\lambda}$ that satisfies the user-specified conservativeness. However, we don't want any of the $\hat{\lambda}_t$'s to be chosen too aggressively, resulting in a threshold too small to be risk-controlled. This is especially important in policy-making, where a policy's risk must be closely monitored as its usage is incrementally increased. Thus, we further require that the trajectory of the $\hat{\lambda}_t$'s is safe for all $t \in [T]$:

$$\mathbb{P}(R(\hat{\lambda}_t) \leq \alpha) \geq 1 - \delta,$$

where $T$ is the iteration of return in our algorithm.

## 3 Performative Risk Control

In this section, we describe PRC's main results. We provide an iterative procedure to compute a series of safe thresholds. Specifically, we demonstrate that our framework provides statistical guarantees on the ability to maintain risk control at user-specified level $\alpha$ throughout the iterative process, as well as obtain $\hat{\lambda}_T$ achieving user-specified tightness at $\alpha - \Delta\alpha$.

---

[2]Our setting follows [5], in which the loss value becomes trivially too large at $\lambda_{\min}$ and too small at $\lambda_{\text{safe}}$. This holds generally in risk control literature.

## 3.1 Our Procedure

To give a high-level summary, we provide an iterative procedure to obtain less conservative thresholds at every iteration. Specifically, until the process is terminated, at time $t \geq 1$, we make progress by choosing $\hat{\lambda}_t$ such that $\hat{\lambda}_t < \hat{\lambda}_{t-1}$. With probability at least $1 - \delta$, our selections remain safe at any time—that is, $R(\hat{\lambda}_t) \leq \alpha$ for all $t \in [T]$—and upon termination, the final selection achieves user-specified tight risk control with $R(\hat{\lambda}_T) \geq \alpha - \Delta\alpha$. PRC starts with setting $\lambda_0 = \lambda_{\text{safe}}$ and then does the following process:

---

**Performative Risk Control (PRC)**

For $t \geq 1$, we sample $\mathcal{I}_{\text{cal}}^t = \{(x_{t-1,i}, y_{t-1,i})\}_{i=1}^{n_{t-1}} \in \mathcal{X}^{n_{t-1}} \times \mathcal{Y}^{n_{t-1}}$ i.i.d. drawn from $\mathcal{D}(\hat{\lambda}_{t-1})$ (here, $\hat{\lambda}_0 = \lambda_0$) and calculate

$$\hat{\lambda}_t = \min\left\{ \inf\{\lambda \in \Lambda \mid V(\hat{\lambda}_{t-1}, \lambda, \delta) \leq \alpha\}, \hat{\lambda}_{t-1} \right\},$$

where $V(\hat{\lambda}_{t-1}, \lambda, \delta)$ is calculated based on the calibration set $\mathcal{I}_{\text{cal}}^t$. If $\hat{\lambda}_t < \hat{\lambda}_{t-1} - \Delta\lambda$ for a progress measure $\Delta\lambda$, which means $\hat{\lambda}_t$ makes enough progress over $\hat{\lambda}_{t-1}$, then continue with $t + 1$. Otherwise, return $\hat{\lambda}_t$.

---

We specify details of $V$ and $\Delta\lambda$ later. Notice that the calibration dataset $\mathcal{I}_{\text{cal}}$ in Def. 1.1 is the union of calibration sets throughout the process, i.e., $\mathcal{I}_{\text{cal}} = \cup_t \mathcal{I}_{\text{cal}}^t$. The formal and complete algorithm is listed in Alg. 1.

## 3.2 Technical Details

In this subsection, we spell out details regarding various parameters and functions in our procedure. Before that, we need to state our key assumption. Intuitively, if the distribution shift of $\mathcal{D}(\lambda)$ between different $\lambda$'s is too dramatic, it will be hard to achieve performative risk control. Thus, we impose our key distributional assumptions below, which is similar to the one proposed in classical performative prediction [7] but more tailored to the risk control setting.

**Definition 3.1** $((\gamma, p, g)$-sensitivity). Let $g : \mathcal{Z} \to \mathbb{R}$ be a deterministic function, and denote $\mathcal{D}_g(\lambda)$ as the induced distribution of $g(z)$ where $z \sim \mathcal{D}(\lambda)$. A distribution map $\mathcal{D}(\cdot)$ is $(\gamma, p, g)$-sensitive if for all $\lambda_1, \lambda_2 \in \Lambda$,

$$W_p(\mathcal{D}_g(\lambda_1), \mathcal{D}_g(\lambda_2)) \leq \gamma|\lambda_1 - \lambda_2|$$

where $W_p$ denotes the Wasserstein-$p$ distance.

We remark here on the similarities and differences between this definition and the definition of $\epsilon$-sensitivity in [7]. First, while $\epsilon$-sensitivity parameterizes distribution shifts through the model parameters $\theta$, $(\gamma, p, g)$-sensitivity does so through the risk-control parameter $\lambda$. Both shifts modify the distribution of samples $z = (x, y)$. In $\epsilon$-sensitivity, $p$ is set to 1, and $\epsilon$ quantifies the magnitude of the shift in the distribution of the samples $z$ themselves. In $(\gamma, p, g)$-sensitivity, $p$ is left to be arbitrary, and $\gamma$ measures how much the distribution of the *transformed samples* $g(z)$ changes as a result of the shift in $z$.

Intuitively, we view $\lambda$ as affecting the distribution of samples $z$, which in turn impacts the loss distribution. The following assumption applies $(\gamma, p, g)$-sensitivity to our setting: the loss distribution arising from samples drawn from similar thresholds are also similar.

**Assumption 3.2** (Lipschitz Distribution Mapping). We assume that $\mathcal{D}(\cdot)$ is $(\gamma, p, \ell(\cdot, \lambda))$-sensitive for all $\lambda \in \Lambda$.

We are now ready to state further technical details. Without loss of generality, we can consider $n_t = n$ for all $t \geq 0$ since we can denote $n$ as the minimum of $\{n_t\}_t$ presented in our procedure, and all the results below still hold.

**Choice of $V$.** We denote $\hat{R}_n(\lambda, \lambda') := \frac{1}{n} \sum_{i=1}^n \ell(z_i(\lambda), \lambda')$ to be the empirical mean on the loss of samples $\{z_i(\lambda)\}_{i=1}^n \overset{\text{i.i.d.}}{\sim} \mathcal{D}(\lambda)$ when evaluated against the threshold $\lambda'$. With this notation, $\mathcal{I}_{\text{cal}}^t$ could also be denoted as $\{z_i(\hat{\lambda}_t)\}_{t=1}^n$. We further denote $R(\lambda, \lambda') := \mathbb{E}_{z \sim \mathcal{D}(\lambda)} \ell(z, \lambda')$.

To construct $V$, we require knowledge of a *confidence width* $c(n, \delta')$ defining pointwise confidence bounds $\hat{R}_n^{\pm}(\lambda, \lambda', \delta') := \hat{R}_n(\lambda, \lambda') \pm c(n, \delta')$ that satisfy the following property for $\hat{\lambda}_{t-1}$ and $\lambda' \geq \hat{\lambda}_t$ for all $t \geq 1$ encountered in the procedure:

$$\mathbb{P}\left(\hat{R}_n^{-}(\hat{\lambda}_{t-1}, \lambda', \delta') \leq R(\hat{\lambda}_{t-1}, \lambda') \leq \hat{R}_n^{+}(\hat{\lambda}_{t-1}, \lambda', \delta')\right) \geq 1 - \delta'. \tag{1}$$

where $\delta' = \delta/\tilde{T}$ and $\tilde{T}$ is an upper bound on the number of iterations $T$. We discuss choices for $c(n, \delta')$–including derivations from Hoeffding's inequality, Bernstein's inequality, Hoeffding-Bentkus's inequality, and the central limit theorem–in App. A.1. We are further often interested in the risk profile of $R(\hat{\lambda}_t, \cdot)$, whose difference from $R(\hat{\lambda}_{t-1}, \cdot)$ can be bounded via sensitivity. We call this difference the *performative error*, or the error that arises from evaluating on two different distributions. In a nutshell, we hope to update $\hat{\lambda}_{t-1} \to \hat{\lambda}_t$ by accounting for increases in the risk due to generalization and performative error, which leads to the following form:

$$V(\hat{\lambda}_{t-1}, \lambda, \delta) := \underbrace{\hat{R}_n(\hat{\lambda}_{t-1}, \lambda)}_{\text{empirical loss}} + \underbrace{c(n, \delta/\tilde{T})}_{\text{confidence width}} + \underbrace{\tau(\hat{\lambda}_{t-1} - \lambda)}_{\text{performativity guard}}.$$

The user-chosen parameter $\tau > 0$ guards against performative effects; we require $\tau \geq \gamma$. While $\gamma$ is typically unknown, one can estimate its potential range or magnitude based on domain knowledge of the problem at hand to then choose a high enough $\tau$.

**Choice of $\Delta\lambda$.** We choose $\Delta\lambda = \frac{1}{2\tau}\left(\Delta\alpha - 2c(n, \delta/\tilde{T})\right)$. Once the update $\hat{\lambda}_{t-1} \to \hat{\lambda}_t$ falls short of the progress measure $\Delta\lambda$, PRC terminates and outputs $\hat{\lambda}_t$.

**Number of iterations $T$ vs. guaranteed convergence $\tilde{T}$.** This distinction is necessary given that per-iteration error rates $\delta'$ compound and $T$ is initially unknown. After $T$ iterations, the trajectory error rate is $\delta'T$, so defining $\delta' = \delta/\tilde{T}$ allows us to bound this error rate at $\delta'T = \delta(T/\tilde{T}) \leq \delta$. While the definition of $\tilde{T}$ increases the confidence width and increases the complexity of PRC, the benefit is that PRC can inform the user whether theoretical guarantees are possible for a given $(\alpha, \Delta\alpha, \delta)$ *prior to any samples being collected.*

**Obtaining $\tilde{T}$ and $\Delta\lambda$.** Since we require $\Delta\lambda$ progress on each iteration on the interval $[\lambda_{\min}, \lambda_{\text{safe}}]$, PRC terminates in at most $\lceil(\lambda_{\text{safe}} - \lambda_{\min})/\Delta\lambda\rceil$ iterations. Hence, choosing $\tilde{T} \geq \lceil(\lambda_{\text{safe}} - \lambda_{\min})/\Delta\lambda\rceil$, or $\Delta\lambda \geq (\lambda_{\text{safe}} - \lambda_{\min})/\tilde{T}$, guarantees convergence, i.e. $T \leq \tilde{T}$, always. Further, because $\Delta\lambda$ is also a function of $\tilde{T}$, we need to jointly solve $(\tilde{T}, \Delta\lambda) \in \mathbb{N} \times \mathbb{R}$. If there are no solutions, PRC cannot guarantee that its sequentially produced solutions are safe; in this case, we just return $\lambda_{\text{safe}}$.

**Details about $\Delta\alpha$.** Suppose we let $\tilde{T}, n \to \infty$ with $\tilde{T} = Cn^r$ (for some constant $C$ and $r \geq 1/2$) and $\Delta\lambda = (\lambda_{\text{safe}} - \lambda_{\min})/\tilde{T}$. Then, $\Delta\alpha = O(\ln(n)/\sqrt{n})$. We leave this derivation to App. A.2.

We summarize all the discussions above with the following detailed and complete algorithm in Alg. 1.

---

**Algorithm 1** Performative Risk Control

---

**Require:** $\alpha$, $\Delta\alpha$, $\delta$, $n$, loss $\ell$, distribution mapping $\mathcal{D}(\cdot)$, $\lambda_{\min}$, $\lambda_{\text{safe}}$, safety parameter $\tau$.
**Ensure:** Output $\hat{\lambda}$
1: Initialize $\lambda_0 \leftarrow \lambda_{\text{safe}}$
2: Jointly solve $(\tilde{T}, \Delta\lambda) \in \mathbb{N} \times \mathbb{R}$ s.t. $2\tau\Delta\lambda = \Delta\alpha - 2c(n, \delta/\tilde{T})$ and $\Delta\lambda \geq (\lambda_{\text{safe}} - \lambda_{\min})/\tilde{T}$.
   Break ties by choosing the solution with the minimum $\tilde{T}$. If there are no solutions, return $\lambda_{\text{safe}}$.
3: **for** $t = 1$ to $\tilde{T}$ **do**
4:     Receive samples $\{z_{t-1,i}\}_{i=1}^n \overset{\text{i.i.d.}}{\sim} \mathcal{D}(\hat{\lambda}_{t-1})$ where $\hat{\lambda}_0 = \lambda_0$
5:     Set $\hat{\lambda}_t \leftarrow \min\left\{\inf\{\lambda \in \Lambda \mid V(\hat{\lambda}_{t-1}, \lambda, \delta) \leq \alpha\}, \hat{\lambda}_{t-1}\right\}$
6:     **if** $\hat{\lambda}_t \geq \hat{\lambda}_{t-1} - \Delta\lambda$ **then**
7:         **return** $\hat{\lambda}_t$
8:     **end if**
9: **end for**

---

## 3.3 Theoretical Guarantees

We now state our main theoretical result. By the following theorem, we demonstrate how our procedure can realize the promise of achieving user-specified conservativeness and being safe and reliable at anytime.

**Theorem 3.3** (Risk control of PRC). *Suppose Assumption 3.2 holds, so that the distribution mapping $\mathcal{D}(\cdot)$ is $(\gamma, 1, \ell(\cdot, \lambda))$-sensitive for all $\lambda \in \Lambda$. Assume further that the loss function $\ell(z, \lambda)$ is continuous in $\lambda$ for all $z$, and that the safety parameter $\tau$ satisfies $\tau \geq \gamma$. If the initial joint solve of $(\tilde{T}, \Delta\lambda)$ produces at least one value, Algorithm 1 guarantees that with probability $1 - \delta$, the following three conditions are met simultaneously:*

> *(i)* **Safety in the iterative process:** $\quad R(\hat{\lambda}_{t-1}, \hat{\lambda}_t) \leq \alpha, \quad 1 \leq t \leq T,$
>
> *(ii)* **Safety at any time:** $\quad R(\hat{\lambda}_t) \leq \alpha, \quad 0 \leq t \leq T,$
>
> *(iii)* **Tightness of $\hat{\lambda}_T$ :** $\quad R(\hat{\lambda}_T) \geq \alpha - \Delta\alpha, \quad T \geq 1.$

**Remark 3.4.** *In Thm. 3.3, we rely on the key distributional assumption 3.2. Because the distribution shift is measured on the loss samples themselves, Alg. 1 accommodates model finetuning (e.g. via supervised finetuning or reinforcement learning) between iterations, as long as assumption 3.2 is satisfied. Consequently, PRC allows data and model to co-evolve, as long as finetuning maintains a relatively stable loss distribution.*

# 4 Extension

In this section, we discuss how PRC can be extended to handle quantile-based risk measures, which is commonly used in quantify tail risk and widely used in mathematical finance. In particular, both conditional variance-at-risk (CVaR) and expected risk belong to quantile-based risk measures. Denote the cumulative distribution function (CDF) of losses for $z \sim \mathcal{D}(\lambda)$ and evaluation threshold $\lambda'$ as $F(w \,;\, \lambda, \lambda') := \mathbb{P}(\ell(z, \lambda') \leq w)$. Further, recall that the inverse of CDF $F$ is defined as $F^{-1}(p) := \inf\{x : F(x) \geq p\}$, which leads to the following definition.

**Definition 4.1** (Quantile-based risk measure). Let $\psi(p)$ be a weighting function such that $\psi(p) \geq 0$ and $\int_0^1 \psi(p)dp = 1$. The quantile-based risk measure defined by $\psi$ is

$$R_\psi(F) = \int_0^1 \psi(p) F^{-1}(p) \, dp$$

From now on, we refer to the quantile-based risk as $R_\psi(\lambda, \lambda') := R_\psi(F(\cdot \,;\, \lambda, \lambda'))$. The analogy to the expected risk case is made clear when $\psi(p) = 1$. Indeed,

$$R_{\psi=1}(\lambda, \lambda') := \int_0^1 F^{-1}(p \,;\, \lambda, \lambda') \, dp = \mathbb{E}_{z \sim \mathcal{D}(\lambda)} \ell(z, \lambda') = R(\lambda, \lambda')$$

Other values of $\psi(p)$ allow us to control the $\beta$-VaR (e.g., the 90th percentile of losses) and $\beta$-CVaR (e.g., the average of the worst $10\%$ of losses). To extend Thm. 3.3, we need to construct a confidence width $c(n, \delta')$. To do so, note that we have knowledge of the empirical loss CDF of the samples from iteration $t - 1$ evaluated against any threshold $\lambda'$:

$$\hat{F}_n(w \,;\, \hat{\lambda}_{t-1}, \lambda') := \frac{1}{n} \sum_{i=1}^n \mathbf{1}\{\ell(z_i(\hat{\lambda}_{t-1}), \lambda') \leq w\}.$$

We follow the technique in [17] and [23] to create an LCB $\hat{F}_{L,n}(\cdot \,;\, \hat{\lambda}_{t-1}, \lambda', \delta')$ and UCB $\hat{F}_{U,n}(\cdot \,;\, \hat{\lambda}_{t-1}, \lambda', \delta')$ based on the empirical CDF, where

$$\mathbb{P}\left( \hat{F}_{L,n}(w \,;\, \hat{\lambda}_{t-1}, \lambda', \delta') \leq F(w \,;\, \hat{\lambda}_{t-1}, \lambda') \leq \hat{F}_{U,n}(w \,;\, \hat{\lambda}_{t-1}, \lambda', \delta') \right) \geq 1 - \delta' \quad \forall w$$

Next, similar to how $R_\psi(\lambda, \lambda')$ is constructed from $F(\cdot\,;\lambda, \lambda')$, we construct $\hat{R}_{\psi,n}^-(\hat{\lambda}_{t-1}, \lambda', \delta')$, $\hat{R}_{\psi,n}(\hat{\lambda}_{t-1}, \lambda')$, and $\hat{R}_{\psi,n}^+(\hat{\lambda}_{t-1}, \lambda', \delta')$ from $\hat{F}_{U,n}(\cdot\,;\hat{\lambda}_{t-1}, \lambda', \delta')$, $\hat{F}_n(\cdot\,;\hat{\lambda}_{t-1}, \lambda', \delta')$, and $\hat{F}_{L,n}(\cdot\,;\hat{\lambda}_{t-1}, \lambda', \delta')$, respectively. Note that the risk measure's lower bound $\hat{R}_{\psi,n}^-(\hat{\lambda}_{t-1}, \lambda', \delta')$ corresponds to the CDF upper bound $\hat{F}_{U,n}(\cdot\,;\hat{\lambda}_{t-1}, \lambda', \delta')$ and vice versa (see App. A.4). The bounds on risk measure $R_\psi(\hat{\lambda}_{t-1}, \lambda')$ satisfy the following:

$$\mathbb{P}\left(\hat{R}_{\psi,n}^-\psi(\hat{\lambda}_{t-1}, \lambda', \delta') \leq R_\psi(\hat{\lambda}_{t-1}, \lambda') \leq \hat{R}_{\psi,n}^+(\hat{\lambda}_{t-1}, \lambda', \delta')\right) \geq 1 - \delta'$$

Let $c(n, \delta', \lambda') = \max\{\hat{R}_{\psi,n}^+(\hat{\lambda}_{t-1}, \lambda', \delta') - \hat{R}_{\psi,n}(\hat{\lambda}_{t-1}, \lambda'), \hat{R}_{\psi,n}(\hat{\lambda}_{t-1}, \lambda') - \hat{R}_{\psi,n}^+(\hat{\lambda}_{t-1}, \lambda', \delta')\}$. Finally, we compute the confidence width $c(n, \delta')$ using the technique described in App. A.1. We are now ready to extend Thm. 3.3 to quantile-based risk measures.

**Theorem 4.2** (Quantile risk control of PRC)**.** *In Alg. 1, replace the confidence width with the one derived above. Further, replace $\hat{R}_n(\hat{\lambda}_{t-1}, \lambda)$ with $\hat{R}_{\psi,n}(\hat{\lambda}_{t-1}, \lambda)$ in the definition of $V$ in line 5. Let $u, v \in [1, \infty]$ with $1/u + 1/v = 1$. If the following four conditions hold: (i) the distribution mapping is $(\gamma, u, \ell(\cdot, \lambda))$-sensitive for all $\lambda \in \Lambda$, (ii) the loss function $\ell(z, \lambda)$ is continuous in $\lambda$ for all $z$, (iii) $\tau \geq \gamma \left[\int_0^1 |\psi(p)|^v dp\right]^{1/v}$, and (iv) the initial joint solve of $(\tilde{T}, \Delta\lambda)$ produces at least one value; then, this modified version of Alg. 1 guarantees that with probability $1 - \delta$, the quantile risk $R_\psi$ of the iterates $\hat{\lambda}$ satisfies the same three guarantees as those in Thm. 3.3.*

## 5 Experiment

This section considers the application of our framework to credit scoring in both the expected and quantile risk settings. We present the most important results here and leave details to App.B.

**Type II Error Control for Credit Scoring.** We first consider type II error control for the binary classification task. Labeling the positive class as $y = 1$, we form the standard prediction $\mathcal{T}_{\lambda,\epsilon}(x) = \text{clip}(\frac{1-\lambda+\epsilon-f(x)}{2\epsilon}, 0, 1)^3$, where $\lambda \in [\lambda_{\min}, \lambda_{\text{safe}}] = [0, 1]$. Here, $f(x)$ estimates the probability that $x$ is positive. We assign $x$ to the negative class when $\mathcal{T}_{\lambda,\epsilon}(x) = 0$ and abstain from making a prediction otherwise. We consider the type II error $\ell(y, \mathcal{T}_{\lambda,\epsilon}(x)) = y(1 - \mathcal{T}_{\lambda,\epsilon}(x))$, accumulating loss for each positive instance assigned to the negative class. We apply this formulation to automatic credit approval. Credit applicants submit their applications with features $x$. The model $f(x)$ predicts the probability of a serious delinquency, and $y$ labels whether one occurred. When $\mathcal{T}_{\lambda,\epsilon}(x) = 0$, the application is automatically accepted; otherwise, it is flagged and subjected to further human review. The threshold $\lambda \in [0, 1]$ trades off the balance of these two, with lower $\lambda$ corresponding to more automatic acceptances. We focus solely on the Type II errors made by the model.

However, $\lambda$ is *performative*. We are not deploying $\lambda$ into a static distribution; applicants respond by maximizing their chances of receiving credit approval. Denote $x(\lambda)$ as an applicant's features in response to the deployment of $\lambda$. We simulate this shift *in the features* by impacting $f(x(\lambda))$ as follows:

$$f(x(\lambda)) = \begin{cases} \max(0, f(x) - s) & \text{if } f(x) - s \leq 1 - \lambda \\ f(x) & \text{otherwise} \end{cases}$$

where $s(= 0.3)$ is a value we choose. This shift is $(\gamma, 1, \ell(\cdot, \lambda))$-sensitive for all $\lambda$ and choice of $s$; we explore the validity of this claim in App. B.1.

We demonstrate results on a class-balanced subset of a Kaggle credit scoring dataset [24], following [7]. Our simulation experiment is reported in Figure 2. We control risk at $\alpha = 0.3$ with $\Delta\alpha = 0.082$. We use $n = 2000$, the CLT bound, and $\delta = 0.1$. We experimentally verify that $\gamma \leq 1.38$ (see App. B.1). As expected from Thm. 3.3, PRC iterates safely as it converges upon a final $\hat{\lambda}$ that is both risk-controlled and not too conservative for large enough $\tau$.

---

[3] We use $\epsilon = 10^{-4}$. This standard prediction approximates the indicator function $\mathbf{1}\{f(x) \leq 1 - \lambda\}$ as $\epsilon \to 0$. For simplicity, our theory assumes a continuous loss function and does not deal with losses with bounded jump discontinuities, hence the approximation here. However, our theory can easily be extended to handle these cases as well, as done in Angelopoulos et al. [6].

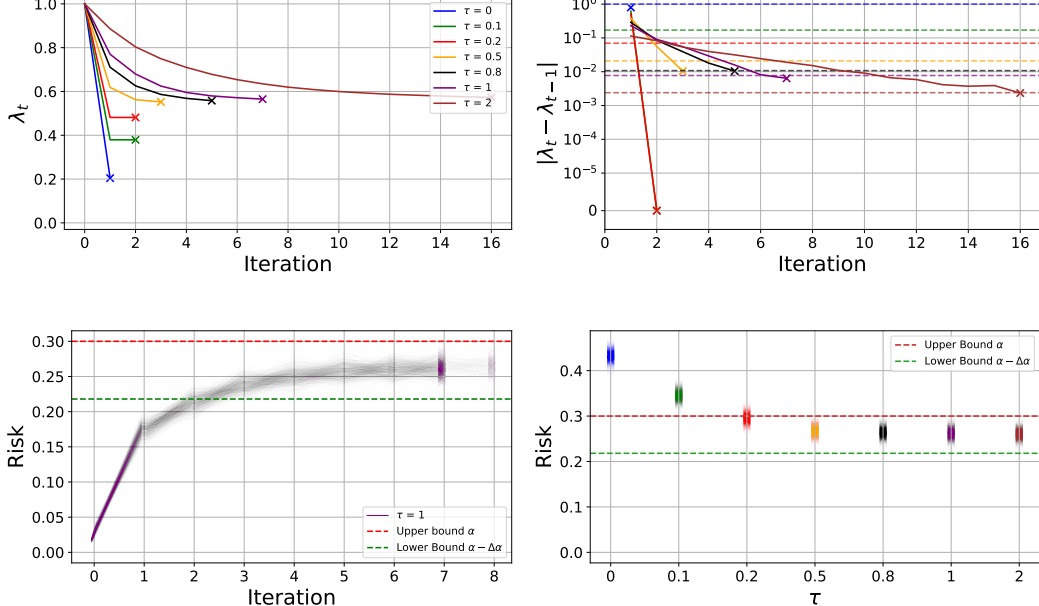

Figure 2: Type II error control in credit scoring. We use the same coloring scheme to label trajectories with different settings of $\tau$ across the subplots. **Top left**: $\tau = 0$ shows an algorithm that does not account for performativity. As $\tau$ increases, the incremental updates to $\hat{\lambda}_t$ become more fine-grained and conservative. This safety comes at the trade-off of longer trajectories. **Top right**: An illustration of our stopping criteria for different values of $\tau$. To guarantee tightness with higher $\tau$, $\Delta\lambda$–displayed as the colored, horizontal dotted lines–needs to be lower. Once $\hat{\lambda}_{t-1} - \hat{\lambda}_t$ falls below $\Delta\lambda$, PRC ends exploration and returns $\hat{\lambda}_T = \lambda_t$. **Bottom left**: $1,000$ trajectories with $\tau = 1$, each trajectory corresponding to a different cut between calibration and validation sets. The iterates $\hat{\lambda}_t$ were chosen based on the calibration set, and the displayed risk is on the validation set. Let $\{z_{t,i} \sim \mathcal{D}(\hat{\lambda}_t)\}_{i=1}^{n_v}$ denote the validation samples from iteration $t$. For each iteration $t$, the plot displays $\frac{1}{n_v}\sum_{i=1}^{n_v} \ell(z_{t,i}, \hat{\lambda}_t)$ followed by $\frac{1}{n_v}\sum_{i=1}^{n_v} \ell(z_{t,i}, \hat{\lambda}_{t+1})$, most noticeable when $t = 0$. Risk is controlled for both quantities. **Bottom right**: A scatterplot of $R(\hat{\lambda}_T)$ for each $\tau$. For $\tau \geq \gamma$, PRC achieves its goal of being $(\alpha, \delta)$-performative-risk-controlled. Even for some values of $\tau < \gamma$ (e.g., $\tau = 0.5, 0.8, 1.0$), the final risk is still controlled (also see App. B.1).

**Quantile Risk Control in Credit Scoring.** In a highly imbalanced dataset where the negative class is much more common than the positive class, risk control on a small expected type II error will not be very meaningful. Quantile-based risk measures target scenarios like this one, in which we seek guarantees on the tail end of negative outcomes. Given that we are investigating quantile-based risk measures, we adjust our setup to more accurately reflect a setting in which they would be useful. For each positive delinquency on each iteration, we simulate a realized cost by drawing from $U[0, 1]$, which can be thought of as the amount the institution loses on an individual, via being late on payments or defaulting, scaled between $0$ and $1$. We also assume knowledge of the base rate $p$ of a credited individual experiencing a serious delinquency; in this experiment, $p = 0.0640$.

We focus on the $\beta$-CVaR risk measure, with $\beta = 0.9$. We use a quantile-based-CLT confidence width that depends on $n$, $\delta$, $\tilde{T}$, $\beta$, and $p$ (see App. B.2). We increase $n$ to $10,000$. Note that we require this many samples because $1 - p$ of the population has a loss of $0$ and is not informative for us in setting $\lambda$. This setting follows Thm 4.2 with $u = 1$, $v = \infty$, and $\gamma \left[\int_0^1 |\psi(p)|^v dp\right]^{1/v} = \gamma/(1 - \beta) \leq \tau$. We experimentally verify $\gamma \leq 0.205$, so Thm. 4.2 applies when $\tau \geq 2.05$. Fig. 3 shows our results for $\alpha = 0.25$ and $\Delta\alpha = 0.12$. We observe that the confidence bounds are somewhat generous; each trajectory ends with a final $R(\hat{\lambda}_T)$ inside the bounds. However, this width is necessary given that this dataset is so highly skewed and that we are controlling the most variable $10\%$ of losses.

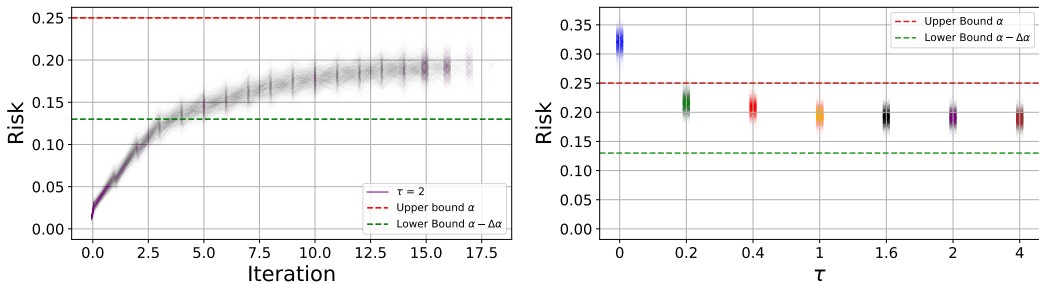

Figure 3: 90%-CVaR type II error control in credit scoring. These plots correspond to the bottom two plots in Fig.2 but for the 90%-CVaR risk measure. The left plot shows $\tau = 2$ instead of $\tau = 1$.

## 6 Conclusion

We have introduced a theoretically sound and empirically effective framework for risk control under performativity, a previously unexplored topic in the risk control literature. To mitigate performativity, our framework iteratively refines a series of decision thresholds, careful to control risk. As a concrete example, we examined the deployment of a credit scoring system for loan approvals and found that PRC successfully controls risk, even as credit applicants seek to strategically manipulate their outcome. One limitation of PRC is sample efficiency, which stems from the iterative nature of performative algorithms and the necessity to form statistical bounds under distribution shift. While we feel that our desiderata of being safety at anytime is important in the performative setting, one could achieve higher sample efficiency by relaxing the desiderata to being safe *on average* over time. Nevertheless, given the growing role of blackbox models in shaping policies of societal significance, PRC offers an effective and proactive strategy for their safe deployment.

**Broader Impacts** This paper proposes a framework to safely and reliably deploy models whose predictions influence society. There are many potential societal consequences of our work, none of which we feel must be specifically highlighted.

## Acknowledgments

We would like to thank Juan Perdomo and Lihua Lei for helpful discussions and feedback. We also thank the anonymous reviewers and area chairs for their insightful comments. QL acknowledges support of NSF DMS-2523382.

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

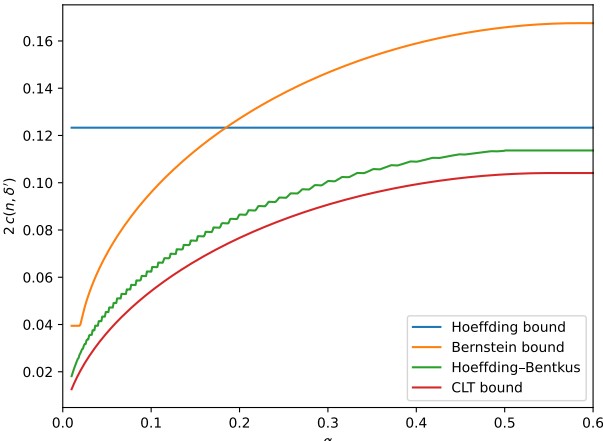

Figure 4: A comparison of Hoeffding and the risk level-dependent bounds. Plots $2c(n, \delta')$ vs. $\alpha$ for $n = 2000$, $\tilde{T} = 100$, $\delta = 0.1$, and $\delta' = \delta/\tilde{T}$. $2c(n, \delta')$ represents the lower limit of achievable tightness $\Delta\alpha$. Lower confidence widths are better.

## A  Proofs

### A.1  Choice of bounds

The purpose of the confidence width $c(n, \delta')$ is to guard against generalization error. Because we need access to $c(n, \delta')$ before the initial deployment and $c(n, \delta')$ must guard against worst-case distributions that could arise from performative shift, the bounds presented here are looser than those found in the literature, yet still fairly tight. In this section, we detail how we can derive confidence widths from a variety of concentration inequalities, a comparison of which is shown in Fig. 4, for a bounded loss scaled in the range $[0, 1]$.

**Hoeffding's inequality**  For any $\lambda, \lambda' \in \Lambda$, if $z \sim \mathcal{D}(\lambda)$, then $\ell(z, \lambda') \in [0, 1]$. Because these are bounded random variables, Hoeffding's inequality applies, which says that for any $\delta' \in (0, 1)$,

$$\mathbb{P}\left(\left|\hat{R}_n(\lambda, \lambda') - R(\lambda, \lambda')\right| \leq c(n, \delta')\right) \geq 1 - \delta'$$

where $c(n, \delta') = \sqrt{\frac{1}{2n} \ln(2/\delta')}$. While we present Hoeffding's inequality for illustration, we use sharper bounds in practice. The rest of the bounds we consider in this section are sample-dependent. We use knowledge of $\alpha$ to calculate their confidence widths, as detailed next.

**Risk level-dependent bounds**  Sample-dependent concentration bounds require some care. Our algorithm calculates the confidence width $c(n, \delta')$ *before the first deployment* without access to the distributions of future deployments. This task of using sample-dependent bounds without initial access to them is further complicated by performative shift, which impedes us from more specific characterizations of these future distributions, and hence from obtaining sharper bounds. However, we have knowledge of the risk control level $\alpha$ and can use that information to minimize $c(n, \delta')$ while making it wide enough to control risk throughout, and at the end of, the iterative process.

Specifically, on iteration $t$, we receive samples from the previous round $\{z_{t-1,i}\}_{i=1}^n \overset{\text{i.i.d.}}{\sim} \mathcal{D}(\hat{\lambda}_{t-1})$ and can calculate $\hat{R}_n(\hat{\lambda}_{t-1}, \lambda)$ for any $\lambda$. $\hat{R}_n(\hat{\lambda}_{t-1}, \lambda)$ is the sample mean loss when evaluating with $\lambda$; from it, we can calculate the maximum sample variance, which occurs when the loss distribution is Bernoulli with probability $\hat{R}_n(\hat{\lambda}_{t-1}, \lambda)$. Since our concentration inequalities use these two quantities to construct their bounds, the exact distribution we draw from does not matter, and the derived confidence widths will work for the distribution *at any round*, regardless of performativity.

Next, note that our algorithm is functionally equivalent if we replace Line 5 of Alg. 1 with the following update rule:

$$\lambda_t^{\mathrm{mid}} \leftarrow \inf\{\lambda \in \Lambda \mid \hat{R}_n^+(\hat{\lambda}_{t-1}, \lambda, \delta') + \tau\max(0, \hat{\lambda}_{t-1} - \lambda) < \alpha\}.$$

Since $\tau\max(0, \hat{\lambda}_{t-1} - \lambda) \geq 0$ always, the candidates $\lambda$ for $\lambda_t^{\mathrm{mid}}$ are a subset of $S_\alpha := \{\lambda \in \Lambda \mid \hat{R}^+(\hat{\lambda}_{t-1}, \lambda, \delta') < \alpha\}$. Assume for all $\lambda$ we have access to a *pointwise* confidence width $c(n, \delta', \lambda)$ that bounds $R(\hat{\lambda}_{t-1}, \lambda)$ by knowledge of the sample statistics calculated from $\hat{R}_n(\hat{\lambda}_{t-1}, \lambda)$ above. Let $\hat{R}_n^\pm(\hat{\lambda}_{t-1}, \lambda, \delta') = \hat{R}_n(\hat{\lambda}_{t-1}, \lambda, \delta') \pm \sup_{\lambda' \geq \lambda} c(n, \delta', \lambda')$. Define the update that does not consider performativity as $\tilde{\lambda}_t^{\mathrm{mid}} := \inf\{S_\alpha\}$. We set the confidence width as

$$c(n, \delta') := \max\{c(n, \delta', \lambda') \mid \lambda' \geq \tilde{\lambda}_t^{\mathrm{mid}}\}.$$

This setting ensures that throughout the entire trajectory of $\hat{\lambda}_t$ for which we bound $\hat{R}_n(\hat{\lambda}_{t-1}\hat{\lambda}_t) + \tau(\hat{\lambda}_{t-1} - \hat{\lambda}_t) \leq \alpha - c(n, \delta')$, we can guarantee with high probability that $R(\hat{\lambda}_{t-1}, \hat{\lambda}_t) + \tau(\hat{\lambda}_{t-1} - \hat{\lambda}_t) \leq \alpha$ also. Next, we use this general result to obtain confidence widths from Bernstein's inequality, the Hoeffding-Benkus inequality, and the central limit theorem.

**Bernstein's inequality**

**Proposition A.1** (Empirical Bernstein Bounds [25, Theorem 11])**.** *Let $\mathbf{X} = (X_1, ..., X_n)$ be a vector of independent random variables with values in $[0, 1]$. Let $\delta > 0$. Then with probability at least $1 - \delta$, we have*

$$\mathbb{E}[P_n(\mathbf{X})] \leq P_n(\mathbf{X}) + \sqrt{\frac{2V_n(\mathbf{X})\ln 2/\delta}{n}} + \frac{7\ln 2/\delta}{3(n-1)}$$

*where $P_n(\mathbf{X})$ denotes the sample mean and $V_n(\mathbf{X})$ denotes the sample variance.*

We observe the sample mean $\hat{R}_n(\hat{\lambda}_{t-1}, \lambda)$ and can upper bound the sample variance with $\hat{R}_n(\hat{\lambda}_{t-1}, \lambda)(1 - \hat{R}_n(\hat{\lambda}_{t-1}, \lambda))$, which is obtained when the losses are Bernoulli. We further convert the one-sided bound into a two-sided bound, giving us the following:

$$c(n, \delta', \lambda) = \sqrt{\frac{2\hat{R}_n(\hat{\lambda}_{t-1}, \lambda)(1 - \hat{R}_n(\hat{\lambda}_{t-1}, \lambda))\ln(4/\delta')}{n}} + \frac{7\ln(4/\delta')}{3(n-1)}.$$

We can then use the technique described above in App. A.1 to obtain the confidence width.

**The Hoeffding-Bentkus inequality**

**Proposition A.2** (Hoeffding-Bentkus inequality p-values [4, Proposition 1])**.** *The following is a valid p-value for $\mathcal{H} : R(\lambda) > \beta$ for $n$ samples*

$$p^{\mathrm{HB}}(n, \hat{R}_n(\lambda), \beta) = \min\left(\exp\left\{-nh_1\left(\hat{R}_n(\lambda) \wedge \beta, \beta\right)\right\}, e\mathbb{P}\left(\mathrm{Bin}(n, \beta) \leq \lceil n\hat{R}_n(\lambda)\rceil\right)\right), \quad (2)$$

*where $R(\lambda)$ is a bounded risk, $\hat{R}_n(\lambda)$ is the sample risk, $h_1(a, b) = a\log(a/b) + (1 - a)\log((1 - a)/(1 - b))$ and $\wedge$ denotes the minimum.*

$R(\hat{\lambda}_{t-1}, \lambda)$ falls in the interval

$$(\hat{R}_n^-(\hat{\lambda}_{t-1}, \lambda, \delta'), \hat{R}_n^+(\hat{\lambda}_{t-1}, \lambda, \delta')) := (\hat{R}_n(\hat{\lambda}_{t-1}, \lambda, \delta') - c, \hat{R}_n(\hat{\lambda}_{t-1}, \lambda, \delta') + c)$$

if the following two hypotheses are rejected:

$$\mathcal{H}^{UB} : R(\hat{\lambda}_{t-1}, \lambda) > \beta^{UB} := \hat{R}_n^+(\hat{\lambda}_{t-1}, \lambda, \delta') \quad \text{with p-value } p^{UB}(n, \hat{R}_n(\hat{\lambda}_{t-1}, \lambda), \beta^{UB}),$$

$$\mathcal{H}^{LB} : 1 - R(\hat{\lambda}_{t-1}, \lambda) > 1 - \beta^{LB} := 1 - \hat{R}_n^-(\hat{\lambda}_{t-1}, \lambda, \delta') \quad \text{with p-value } p^{LB}(n, 1 - \hat{R}_n(\hat{\lambda}_{t-1}, \lambda), 1 - \beta^{LB}).$$

We simply choose the narrowest confidence width that provides us with $\delta'$ error control:

$$c(n, \delta', \lambda) := \arg\min_{c \geq 0} p^{UB}(n, \hat{R}_n(\hat{\lambda}_{t-1}, \lambda), \beta^{UB}) + p^{LB}(n, 1 - \hat{R}_n(1 - \hat{\lambda}_{t-1}, \lambda), 1 - \beta^{LB}) \leq \delta'.$$

**Central limit theorem**    Given $\hat{R}_n(\lambda_{t-1}, \lambda)$, the maximum sample variance is

$$\hat{\sigma}^2(n, \lambda) = \frac{1}{n-1} \sum_{i=1}^{n} \left( \ell(z_{t-1}^{(i)}, \lambda) - \hat{R}_n(\lambda_{t-1}, \lambda) \right)^2.$$

Denote the cumulative distribution function (CDF) of the standard normal as $\Phi$. With probability $1 - \delta'$, $|R(\lambda_{t-1}, \lambda) - \hat{R}_n(\lambda_{t-1}, \lambda) \leq c(n, \delta', \lambda)$ where $c(n, \delta', \lambda) = \Phi^{-1}(1 - \delta'/2)\hat{\sigma}(\lambda)/\sqrt{n}$.

## A.2    Asymptotic results for the lower bound

Recall that $\Delta\alpha = 2\tau\Delta\lambda + 2c(n, \delta/\tilde{T})$ where $\tilde{T} \geq \lceil(\lambda_{\text{safe}} - \lambda_{\text{min}})/\Delta\lambda\rceil$. First, we need to ensure that there exist solutions $(\tilde{T}, \Delta\lambda)$ in this asymptotic regime. For a given $\Delta\lambda$, we can choose the minimum $\tilde{T}$ subject to its constraint and let $n \to \infty$. Doing so allows $c(n, \delta/\tilde{T}) \to 0$. Since $\Delta\lambda$ can be chosen arbitrarily small, we can make $\Delta\alpha$ go to zero.

Suppose we let $\Delta\lambda \to 0$ and scale the number of samples $n$ as $\Delta\lambda = Cn^{-r}$ (for some constant $C$ and sufficiently high $r \geq 1/2$). We remark that the confidence width for the bounds we consider can be expressed asymptotically as $c(n, \delta/\tilde{T}) = O(\sqrt{\frac{\ln(\tilde{T}/\delta)}{n}}) = O(\sqrt{\frac{\ln(1/\Delta\lambda)}{n}}) = O(\frac{r\ln(n)}{\sqrt{n}}) = O(\frac{\ln n}{\sqrt{n}})$. Hence, asymptotically

$$\Delta\alpha = 2\tau\Delta\lambda + 2c(n, \delta/\tilde{T})$$
$$= O(n^{-r}) + O(\frac{\ln n}{\sqrt{n}})$$
$$= O(\frac{\ln n}{\sqrt{n}})$$

for $r \geq 1/2$. PRC allows users to choose a vanishingly tight lower bound $\Delta\alpha \to 0$.

## A.3    Main proofs

In this section, we build towards our final goal of proving Thm. 3.3. We do so by first showing that the performative error, arising from risk measures with the same evaluation threshold but whose samples are drawn from different distributions, can be bounded. We then demonstrate the validity of UCB calibration to bound the generalization error. Taken together, we can then prove Thm. 3.3.

The following lemma uses the definition of a quantile-based risk measure from Def. 4.1. Notably, $\psi(p)$ is a weighting function; $\psi(p) = 1$ when we consider expected risk. It is important to remark that when $\psi(p) = 1$, $\left[\int_0^1 |\psi(p)|^u dp\right]^{1/u} = 1$ for $u \in [1, \infty]$.

**Lemma A.3.** *Let* $u, v \in [1, \infty]$ *with* $1/u + 1/v = 1$. *If* $\left[\int_0^1 |\psi(p)|^u dp\right]^{1/u} \leq M$ *and* $\mathcal{D}(\cdot)$ *is* $(\gamma, v, \ell(\cdot, \lambda))$-*sensitive for all* $\lambda \in \Lambda$, *then for any* $\lambda_1, \lambda_2, \lambda \in \Lambda$:

$$|R_\psi(\lambda_1, \lambda) - R_\psi(\lambda_2, \lambda)| \leq \gamma M|\lambda_1 - \lambda_2|.$$

*Proof.*

$$\begin{aligned}
|R_\psi(\lambda_1, \lambda) - R_\psi(\lambda_2, \lambda)| &= \left| \int_0^1 \psi(p) F^{-1}(\cdot\,;\lambda_1, \lambda) dp - \int_0^1 \psi(p) F^{-1}(\cdot\,;\lambda_2, \lambda) dp \right| \\
&= \left| \int_0^1 \psi(p) \left[ F^{-1}(\cdot\,;\lambda_1, \lambda) - F^{-1}(\cdot\,;\lambda_2, \lambda) \right] dp \right| \\
&= \int_0^1 \psi(p) \left| F^{-1}(\cdot\,;\lambda_1, \lambda) - F^{-1}(\cdot\,;\lambda_2, \lambda) \right| dp \\
&= \left( \int_0^1 |\psi(p)|^u dp \right)^{1/u} \left( \int_0^1 \left| F^{-1}(p\,;\lambda_1, \lambda) - F^{-1}(p\,;\lambda_2, \lambda) \right|^v dp \right)^{1/v} \\
&\leq M \left( \int_0^1 \left| F^{-1}(p\,;\lambda_1, \lambda) - F^{-1}(p\,;\lambda_2, \lambda) \right|^v dp \right)^{1/v} \\
&= M W_v(\mathcal{D}(\lambda_1), \mathcal{D}(\lambda_2)) \\
&\leq M\gamma|\lambda_1 - \lambda_2|
\end{aligned}$$

$\square$

As a note on notation, from now on until the rest of this proof section, we write the risk measure as $R$, understanding that this refers to the more general form $R_\psi$ (where the subscript $\psi$ has been dropped).

In the following proposition, we restate Theorem A.1 in Bates et al. [5] in our performative setting. This theorem is key to turning the pointwise bound used in line 5 of Algorithm 1 into a proof of the validity of each iterate $\hat{\lambda}$.

**Proposition A.4** (Validity of UCB Calibration)**.** *Assume we have deployed some $\lambda \in \Lambda$ to obtain samples from the distribution $\mathcal{D}(\lambda)$. Let the true risk function $R(\lambda, \cdot) : \Lambda \to \mathbb{R}$ and its $1 - \delta'$ upper confidence bound (UCB) $\hat{R}_n^+(\lambda, \cdot\,; \delta') : \Lambda \to \mathbb{R}$ be continuous and non-increasing functions. Suppose there exists some $\lambda' \in \Lambda$ such that both*

$$R(\lambda, \lambda') \leq \alpha \quad \text{and} \quad \hat{R}_n^+(\lambda, \lambda'; \delta') \leq \alpha.$$

*Consider the smallest $\lambda_*^+$ whose UCB falls below the risk level:*

$$\lambda_*^+ := \inf\{\lambda' \in \Lambda : \hat{R}_n^+(\lambda, \lambda', \delta') < \alpha\}$$

*Then,*

$$\mathbb{P}\left(R(\lambda, \lambda_*^+) \leq \alpha\right) \geq 1 - \delta'$$

*Proof.* First, we claim that $\hat{R}_n^+(\lambda, \cdot, \delta') := \hat{R}_n(\lambda, \cdot) + c(n, \delta')$ is non-increasing and continuous. By the property of the loss function, $\hat{R}_n(\lambda, \cdot)$ is non-increasing and continuous. Further, $c(n, \delta')$ is constant by construction, proving the claim.

Assume $R(\lambda, \lambda_*^+) > \alpha$. Consider the smallest $\lambda_*$ that controls the risk

$$\lambda_* := \inf\{\lambda' \in \Lambda : R(\lambda, \lambda') \leq \alpha\}$$

Because $R(\lambda, \cdot)$ is non-increasing and continuous, $\lambda_*^+ < \lambda_*$. Further, since $\lambda_*^+$ satisfies $\hat{R}_n^+(\lambda, \lambda_*^+, \delta') < \alpha$ and $\hat{R}_n^+(\lambda, \cdot, \delta')$ is also non-increasing and continuous, $\hat{R}_n^+(\lambda, \lambda_*, \delta') < \alpha$. However, since $R(\lambda, \lambda_*) = \alpha$ (by continuity), the pointwise confidence bounds in Equation 1 ensures that this occurs with probability at most $1 - \delta'$. $\square$

Now that we have established how we can bound the performative and generalization error, we now prove the following important theorem, which can then be used to prove each point of Thm. 3.3.

**Theorem A.5** (Safety of Each Deployment)**.** *Take the setting of Theorem 3.3. With probability $1 - \delta$, $R(\hat{\lambda}_t, \hat{\lambda}_t) \leq \alpha$ is ensured for all $t$ in $0 \leq t \leq T$ (here, we denote $\hat{\lambda}_0 = \lambda_0$).*

*Proof.* We establish this theorem by verifying the following inductive statement: on iteration $t$, $R(\hat{\lambda}_s) \leq \alpha$ for all $s$ in $0 \leq s \leq t$ with probability $1 - \delta t / \tilde{T}$.

To establish the base case, recall the definition of $\lambda_{\text{safe}}$, from which we see $\ell(Z, \lambda_0) = \ell(Z, \lambda_{\text{safe}}) = 0 < \alpha$, trivially satisfying $R(\lambda_0) < \alpha$ with probability 1. Afterwards, for $t \geq 1$, there are two cases:

**Case 1:** $\hat{\lambda}_t = \hat{\lambda}_{t-1}$. From induction, $R(\hat{\lambda}_s) < \alpha$ for all $s$ in $0 \leq s \leq t - 1$ with probability $1 - \delta(t - 1)/\tilde{T}$. Since $\hat{\lambda}_t = \hat{\lambda}_{t-1}$, $R(\hat{\lambda}_s) < \alpha$ for all $s$ in $0 \leq s \leq t$ with probability $1 - \delta t / \tilde{T} < 1 - \delta(t - 1)/\tilde{T}$.

**Case 2:** $\hat{\lambda}_t < \hat{\lambda}_{t-1}$. This implies $\hat{\lambda}_t = \inf\{\lambda \in \Lambda \mid V(\hat{\lambda}_{t-1}, \lambda, \delta) \leq \alpha\}$. The following chain of inequalities holds with probability $1 - \delta / \tilde{T}$:

$$
\begin{aligned}
R(\hat{\lambda}_t) &= R(\hat{\lambda}_{t-1}, \hat{\lambda}_t) + R(\hat{\lambda}_t, \hat{\lambda}_t) - R(\hat{\lambda}_{t-1}, \hat{\lambda}_t) \\
&\leq R(\hat{\lambda}_{t-1}, \hat{\lambda}_t) + \gamma M |\hat{\lambda}_t - \hat{\lambda}_{t-1}| \\
&\leq \hat{R}_n^+(\hat{\lambda}_{t-1}, \hat{\lambda}_t, \delta / \tilde{T}) + \gamma M |\hat{\lambda}_t - \hat{\lambda}_{t-1}| \\
&= \alpha - \tau(\hat{\lambda}_{t-1} - \hat{\lambda}_t) + \gamma M |\hat{\lambda}_t - \hat{\lambda}_{t-1}| \\
&= \alpha - (\tau - \gamma M) |\hat{\lambda}_t - \hat{\lambda}_{t-1}| \\
&\leq \alpha
\end{aligned}
$$

where the second line follows from Lemma A.3, the third line from Proposition A.4, and the fourth line follows from the continuity of the loss function $\ell(z, \lambda)$ in $\lambda$ for all $z$.

Since by induction $R(\hat{\lambda}_s) < \alpha$ for all $s$ in $0 \leq s \leq t - 1$ with probability $1 - \delta(t - 1)/\tilde{T}$, taking the union bound verifies that risk is control for all $s$ in $0 \leq s \leq t$ with probability $1 - \delta t / \tilde{T}$. Since $T \leq \tilde{T}$, this completes the proof. □

Having established Thm. A.5, we can now prove each part of Thm. 3.3, which we do in the following three proofs. Note that we denote $M = \left[\int_0^1 |\psi(p)|^u dp\right]^{1/u}$ as in Lemma A.3, and $M = 1$ for the case of expected risk.

*Proof of part (i).* We verify the following inductive statement: with probability at least $1 - \delta t / \tilde{T}$, $R(\hat{\lambda}_{s-1}, \hat{\lambda}_s) \leq \alpha$ for all $s$ in $0 < s \leq t$. Once verified, since PRC takes at most $\tilde{T}$ iterations to finish, the theorem follows.

**Base case:** $t = 1$

If $\hat{\lambda}_1 = \lambda_0$, then with probability 1, $R(\lambda_0, \hat{\lambda}_1) = 0 < \alpha$ by the definition of $\lambda_{\text{safe}}$. Otherwise, $\hat{\lambda}_1 = \inf\{\lambda \in \Lambda \mid V(\lambda_0, \lambda, \delta) \leq \alpha\}$, and with probability $1 - \delta / \tilde{T}$,

$$
\begin{aligned}
R(\lambda_0, \hat{\lambda}_1) &\leq \hat{R}_n^+(\lambda_0, \hat{\lambda}_1, \delta / \tilde{T}) \\
&= \alpha - \tau M(\lambda_0 - \hat{\lambda}_1) \\
&\leq \alpha
\end{aligned}
$$

The first inequality follows from Proposition A.4, and the last follows because if $\hat{\lambda}_1 = \inf\{\lambda \in \Lambda \mid V(\hat{\lambda}_{t-1}, \lambda, \delta) \leq \alpha\}$, then $\hat{\lambda}_1 < \lambda_0$.

**Inductive step:** $t > 1$

Assume $R(\hat{\lambda}_{s-1}, \hat{\lambda}_s) \leq \alpha$ for all $s$ in $0 < s \leq t - 1$ with probability $1 - \delta(t - 1)/\tilde{T}$. On iteration $t$, either $\hat{\lambda}_t = \hat{\lambda}_{t-1}$ or $\hat{\lambda}_t < \hat{\lambda}_{t-1}$. If $\hat{\lambda}_t = \hat{\lambda}_{t-1}$, we appeal to Thm. A.5 to ensure that $R(\hat{\lambda}_{t-1}, \hat{\lambda}_t) \leq \alpha$ with probability $1 - \delta t / \tilde{T}$. The error rate of $\delta(t - 1)/\tilde{T}$ from the inductive step does not accumulate with that of Theorem A.5 because both rely on the same pointwise UCB $\hat{R}_n^+(\hat{\lambda}_{t-1}, \cdot)$ per iteration in the same trajectory. Hence, their maximum becomes the new error rate $\delta t / \tilde{T}$.

Otherwise, $\hat{\lambda}_t < \hat{\lambda}_{t-1}$. In this case, we follow similar steps to the base case. With probability $1 - \delta/\tilde{T}$,

$$
\begin{aligned}
R(\hat{\lambda}_{t-1}, \hat{\lambda}_t) &\leq \hat{R}_n^+(\hat{\lambda}_{t-1}, \hat{\lambda}_t, \delta/\tilde{T}) \\
&= \alpha - \tau M(\hat{\lambda}_{t-1} - \hat{\lambda}_t) \\
&\leq \alpha
\end{aligned}
$$

Taking the union bound with the inductive assumption, we complete the proof. $\square$

*Proof of part (ii).* Appealing to Thm. A.5 completes this part of the proof.

$\square$

*Proof of part (iii).* Algorithm 1 returns $\hat{\lambda}_T$ as the final risk-controlling conservativeness parameter. From Equation 1, with probability $1 - \delta' = 1 - \delta/\tilde{T}$,

$$
\hat{R}_n(\hat{\lambda}_{T-1}, \hat{\lambda}_T) - c(n, \delta') \leq R(\hat{\lambda}_{T-1}, \hat{\lambda}_T) \leq \hat{R}_n(\hat{\lambda}_{T-1}, \hat{\lambda}_T) + c(n, \delta').
$$

There are two cases to consider at the algorithm's end.

**Case 1:** $\hat{\lambda}_T = \hat{\lambda}_{T-1}$. This implies $\hat{\lambda}_{T-1} \leq \inf\{\lambda \in \Lambda \mid V(\hat{\lambda}_{t-1}, \lambda, \delta) \leq \alpha\} := \hat{\lambda}_T^{\mathrm{mid}}$.

$$
\begin{aligned}
R(\hat{\lambda}_T) &= R(\hat{\lambda}_{T-1}) \\
&\geq \hat{R}_n^-(\hat{\lambda}_{T-1}, \hat{\lambda}_{T-1}, \delta/\tilde{T}) \\
&= \hat{R}_n^+(\hat{\lambda}_{T-1}, \hat{\lambda}_{T-1}, \delta/\tilde{T}) - 2c(n, \delta/\tilde{T}) \\
&\geq \hat{R}_n^+(\hat{\lambda}_{T-1}, \hat{\lambda}_T^{\mathrm{mid}}, \delta/\tilde{T}) - 2c(n, \delta/\tilde{T}) \\
&= \alpha - \tau(\hat{\lambda}_{T-1} - \hat{\lambda}_T^{\mathrm{mid}}) - 2c(n, \delta/\tilde{T}) \\
&\geq \alpha - 2c(n, \delta/\tilde{T})
\end{aligned}
$$

where the first inequality is with probability $1 - \delta/\tilde{T}$ and the second inequality follows from the continuity and monotonicity of $\hat{R}_n(\lambda, \cdot)$ for all $\lambda$.

**Case 2:** $\hat{\lambda}_{T-1} - \Delta\lambda \leq \hat{\lambda}_T < \hat{\lambda}_{T-1}$. Recall that by Lemma A.3, we can bound the performative error:

$$
\left| R(\hat{\lambda}_T, \hat{\lambda}_T) - R(\hat{\lambda}_{T-1}, \hat{\lambda}_T) \right| \leq \gamma M |\hat{\lambda}_T - \hat{\lambda}_{T-1}|.
$$

With probability $1 - \delta' = 1 - \delta/\tilde{T}$,

$$
\begin{aligned}
R(\hat{\lambda}_T) &= R(\hat{\lambda}_{T-1}, \hat{\lambda}_T) + \left[ R(\hat{\lambda}_T, \hat{\lambda}_T) - R(\hat{\lambda}_{T-1}, \hat{\lambda}_T) \right] \\
&\geq R(\hat{\lambda}_{T-1}, \hat{\lambda}_T) - \gamma M |\hat{\lambda}_{T-1} - \hat{\lambda}_T| \\
&\geq \hat{R}_n^-(\hat{\lambda}_{T-1}, \hat{\lambda}_T, \delta') - \gamma M |\hat{\lambda}_{T-1} - \hat{\lambda}_T| \\
&= \hat{R}_n^+(\hat{\lambda}_{T-1}, \hat{\lambda}_T, \delta') - \gamma M |\hat{\lambda}_{T-1} - \hat{\lambda}_T| - 2c(n, \delta') \\
&= \alpha - \tau(\hat{\lambda}_{T-1} - \hat{\lambda}_T) - \gamma M |\hat{\lambda}_{T-1} - \hat{\lambda}_T| - 2c(n, \delta') \\
&\geq \alpha - 2\tau(\hat{\lambda}_{T-1} - \hat{\lambda}_T) - 2c(n, \delta') \\
&\geq \alpha - 2\tau\Delta\lambda - 2c(n, \delta') \\
&= \alpha - \Delta\alpha
\end{aligned}
$$

where we set $\Delta\lambda$ such that $\Delta\alpha = 2\tau\Delta\lambda + 2c(n, \delta')$. Doing so allows us to achieve $\Delta\alpha$ tightness with probability $1 - \delta$.

$\square$

## A.4 Quantile extension proofs

**The risk measure's lower bound corresponds to the CDF upper bound.** Assume we have access to lower and upper CDF confidence bounds such that for all $\lambda \in \{\hat{\lambda}_t\}_{t=0}^T$ and $\lambda' \in \Lambda$,

$$\mathbb{P}\left(\hat{F}_{L,n}(w\,;\lambda, \lambda', \delta') \leq F(w\,;\lambda, \lambda') \leq \hat{F}_{U,n}(w\,;\lambda, \lambda', \delta')\right) \geq 1 - \delta' \quad \forall w$$

We rewrite the above as $\mathbb{P}\left(\hat{F}_{L,n} \preceq F \preceq \hat{F}_{U,n}\right) \geq 1 - \delta'$, dropping the $\lambda$ and $\lambda'$ for the rest of this section. The following property establishes that $\hat{F}_{L,n} \preceq F \preceq \hat{F}_{U,n}$ implies $\hat{F}_{U,n}^{-1} \preceq F^{-1} \preceq \hat{F}_{L,n}^{-1}$.

**Proposition A.6.** *If $F \preceq G$, then $F^{-1} \succeq G^{-1}$.*

*Proof.* Recall the generalized inverse of a CDF $H$ is

$$H^{-1}(p) = \inf\{x \in \mathbb{R} : H(x) \geq p\}, \quad p \in (0, 1).$$

Since $F(x) \leq G(x)$ for all $x$, we have $\{x : G(x) \geq u\} \subseteq \{x : F(x) \geq u\}$, so

$$F^{-1}(u) = \inf\{x : F(x) \geq u\} \geq \inf\{x : G(x) \geq u\} = G^{-1}(u).$$

Hence $F^{-1} \succeq G^{-1}$. $\square$

Applying a weighting function does not change this relationship. Hence,

$$\mathbb{P}\left(\hat{R}_{\psi,n}^-\psi(\hat{\lambda}_{t-1}, \lambda', \delta') \leq R_\psi(\hat{\lambda}_{t-1}, \lambda') \leq \hat{R}_{\psi,n}^+(\hat{\lambda}_{t-1}, \lambda', \delta')\right) \geq 1 - \delta'$$

as desired.

*Proof of Theorem 4.2.* This follows as a consequence of the proofs in App. A.3. $\square$

## B  Experiments

In this section, we further detail our experimental setup and include other supporting results. Our credit scoring simulations were conducted on a CPU and can be finished in a few hours on a standard laptop computer.

### B.1  Type II Error Control for Credit Scoring

**Set up and $\gamma$-sensitivity**  Our data is sourced from the training split of a Kaggle credit scoring dataset [24], which contains roughly $150k$ data points, $10k$ of which ending up in a serious delinquency ($y = 1$). We first sample $1.5k$ from each class to train a logistic classifier to function as our model $f(x)$. Because the rest of this dataset is so highly imbalanced, the expected loss would be close to $0$. For illustration purposes, we sample $8.5k$ instances from the negative class so that the proportion of positive and negative classes are balanced. Then, for each trajectory, we sample $n$ points without replacement to serve as the calibration set, with the remaining serving as the validation set on which the risk is evaluated on. To generate multiple trajectories, we sample different calibration and validation sets from the balanced subset.

We now prove that our simulated shift is $(\gamma, 1, \ell(\cdot, \lambda))$-sensitive for all $\lambda$. Recall that our shift is defined as

$$f(x(\lambda)) = \begin{cases} \max(0, f(x) - s) & \text{if } f(x) - s \leq 1 - \lambda \\ f(x) & \text{otherwise.} \end{cases}$$

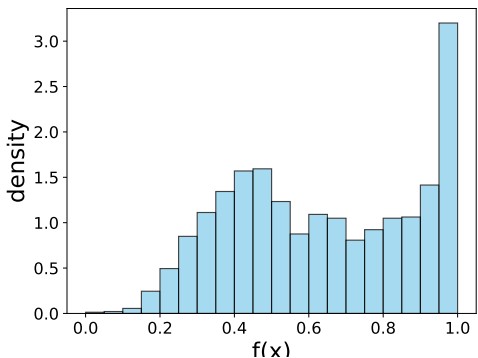

Figure 5: Histogram of predictions $f(x)$ on the base distribution.

where $s(= 0.3)$ is an arbitrary parameter we define.

Consider $\lambda_1$ and $\lambda_2$ so that $1 - \lambda_1 \leq 1 - \lambda_2 \leq 1 - \lambda_1 + s$. Then, $f(x(\lambda_1)$ and $f(x(\lambda_2))$ are equivalent except in the narrow region $[1 - \lambda_1 + s, 1 - \lambda_2 + s]$ (here, we assume we start with the same $x$ and that the region is inside $[0, 1]$). Let $p$ be the probability of a positive case, and let $C$ be the maximum value of the PDF of predictions $f(x)$ on the base distribution given $y = 1$. It is possible that the threshold $\lambda$ causes a discrepancy of $1$ in the loss for these data points in the population by flipping the prediction, e.g., $\lambda = \lambda_1$. Hence, $W_1(\mathcal{D}_{\ell(\cdot,\lambda)}(\lambda_1), \mathcal{D}_{\ell(\cdot,\lambda)}) \leq pC|\lambda_1 - \lambda_2|$ for all $\lambda$.

Now consider when $\lambda_1 + s < \lambda_2$. If we use a threshold $\lambda$ so that $\lambda_1 + s \leq \lambda \leq \lambda_2$, the original loss distribution and that induced by $\lambda_1$ are the same. However, for the distribution induced by $\lambda_2$, the original predictions falling in the region $[1 - \lambda, 1 - \lambda + s]$ flip from 0 to 1. Similar to above, this implies that $W_1(\mathcal{D}_{\ell(\cdot,\lambda)}(\lambda_1), \mathcal{D}_{\ell(\cdot,\lambda)}) \leq pCs$ for all $\lambda$.

Combining the two cases above, we have $W_1(\mathcal{D}_{\ell(\cdot,\lambda)}(\lambda_1), \mathcal{D}_{\ell(\cdot,\lambda)}) \leq pC\min(s, |\lambda_1 - \lambda_2|) \leq pC|\lambda_1 - \lambda_2|$ for all $\lambda$, achieving $\gamma$-sensitivity with $\gamma = pC$. We experimentally verify the values of $C$ and $p$ to be 3.20 and 0.432, respectively. Importantly, we calculate $C$ using the balanced subset rather than the training set. We use 20 bins for the histogram (see Fig. 5). Hence, $\gamma = pC \leq 1.38$.

**Further experimental discussion**  Here, we include supporting plots to Fig. 2 and further discuss our experimental results. Fig. 6 shows the proportion of runs with $\alpha - \Delta\alpha \leq R(\hat{\lambda}_T) \leq \alpha$, and Fig. 7 displays similar plots to the bottom left plot of Fig. 2 for different values of $\tau$.

Fig. 6 plots the failure probability for different values of $\tau$, defined as the proportion of trajectories where the final $\hat{\lambda}_T$ does not satisfy $\alpha - \Delta\alpha \leq R(\hat{\lambda}_T) \leq \alpha$. For $\tau = 1$ and $\tau = 2$ close to $\gamma \leq 1.38$, the failure probability is essentially zero (much less than $\delta$), which some may argue constitutes loose coverage. While it is true that PRC's tightness guarantees are looser than those prevalent in the risk control literature, the unique setting of applying risk control under performative shift warrants their size, which is driven by three main factors. One, our algorithm needs to protect *every iteration* for unknown functions $R(\lambda)$. The risk could be adversarial and necessitate many iterations, all of which PRC needs to protect. Two, our calculation of $\gamma$ is a uniform bound across the iterations, necessitating a larger safety parameter $\tau$; for example, it is possible that the "true" $\gamma$ is between $\tau = 0.2$ and $\tau = 0.5$, at which point the empirical failure rate crosses $\delta$. With the other parameters such as the number of samples $n$ held constant, a larger $\tau$ corresponds to a larger $\Delta\alpha$. Finally, we do not assume "insider" access to the statistics of the first or future distributions. We plan for the worst-case in terms of sample mean and sample variance in computing the confidence widths. These factors contribute to the size of the bounds.

## B.2   Quantile Risk Control in Credit Scoring

**Set up, $\gamma$-sensitivity, and the confidence width**  For this experiment, we adopt the same setting as in expected risk and use the same shift technique. However, after training the logistic classifier, *we*

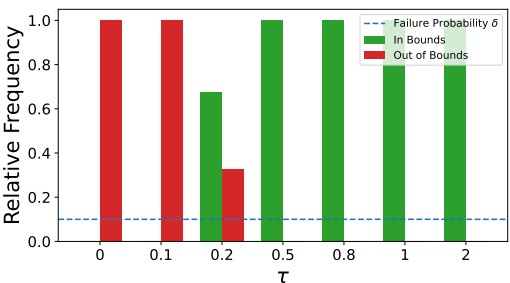

Figure 6: The relative frequency of runs that end with a $\hat{\lambda}_T$ with $\alpha - \Delta\alpha \leq R(\hat{\lambda}_T) \leq \alpha$ for risk control in credit scoring.

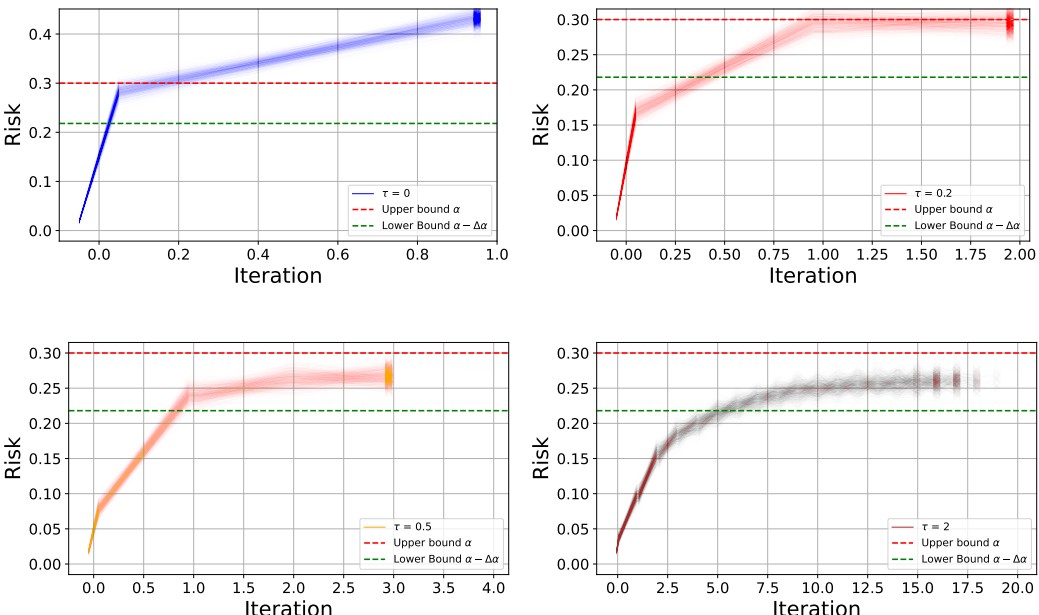

Figure 7: Analog of the bottom left plot in Fig. 2 for $\tau \in \{0, 0.2, 0.5, 2\}$.

*avoid any subsampling,* using the rest of the dataset. Hence, while $C$ remains the same, $p$ drops from 0.432 to 0.064, reducing $\gamma$ to $\gamma = pC \leq 0.205$.

To calculate the confidence width, we follow a recipe based on a version of CLT that is applicable to quantile-based risk measures. The recipe calls for knowledge of the loss CDF. Recall that our loss is given by $\ell(y, \mathcal{T}_\lambda(x)) = yU[0,1](1 - \mathcal{T}_{\lambda,\epsilon}(x))$; we make a minor approximation to this loss function by taking $\epsilon \to 0$, so that the loss is binary. Denote by $p'$ the proportion of the population that has $y(1 - \mathcal{T}_{\lambda,0}(x)) = 1$. Because at most $p$ of the population has $y = 1$, we know that $p' \leq p$, which holds for any given distribution-inducing $\lambda$ and threshold $\lambda'$. Knowing $p'$ allows us to derive the CDF and its inverse, which are piecewise linear because of the uniform distribution.

$$F(w) = \begin{cases} 0 & \text{when } w < 0 \\ 1 - p' + p'w & \text{when } 0 \leq w \leq 1 \\ 1 & \text{when } w > 1 \end{cases}$$

$$F^{-1}(r) = \begin{cases} 0 & \text{when } 0 < r \leq 1 - p \\ 1 - \frac{1-r}{p} & \text{when } 1 - p < r \leq 1 \end{cases}$$

By Theorem 22.3 in Vaart [26], we have

$$\sqrt{n}(R_\psi(\hat{F}_n) - R_\psi(F)) \to_D N(0, \sigma^2(\psi, F)),$$

where

$$\sigma^2(\psi, F) = \iint (F(r \wedge \tilde{r}) - F(r)F(\tilde{r}))\psi(F(r))\psi(F(\tilde{r})) \, dr \, d\tilde{r}.$$

Since the $\beta$-CVaR metric is bounded everywhere and has one discontinuous point, we can apply this theorem (appealing to dominated convergence). We split the calculation of $\sigma^2(\psi, F)$ into two cases.

**Case I:** $\beta \leq 1 - p'$

$$\sigma^2(\psi, F) = \frac{1}{(1-\beta)^2} \int_0^\infty \int_0^\infty F(r \wedge \tilde{r}) - F(r)F(\tilde{r})drd\tilde{r}$$

$$= \frac{1}{(1-\beta)^2} \int_0^1 \int_0^1 F(r \wedge \tilde{r}) - F(r)F(\tilde{r})drd\tilde{r}$$

Using $F(r) = 1 - p' + p'r$ gives us

$$\sigma^2(\psi, F) = \frac{1}{(1-\beta)^2} \frac{1}{12}(4 - 3p')p'$$

**Case II:** $\beta > 1 - p'$    We require $F(r) = 1 - p' + p'r \geq \beta$, or $r \geq (\beta - 1 + p)/p$ for the weighting function to be nonzero. Similarly to above, the upper bound can be changed from $\infty$ to 1 because $F(r) = 1$ for $r \geq 1$. The lower bound is then changed to $a = 1 - \frac{1-\beta}{p}$.

$$\sigma^2(\psi, F) = \frac{1}{(1-\beta)^2} \int_a^1 \int_a^1 F(r \wedge \tilde{r}) - F(r)F(\tilde{r})drd\tilde{r}$$

$$= \frac{2}{(1-\beta)^2} \int_a^1 \int_r^1 F(r) - F(r)F(\tilde{r})drd\tilde{r}$$

$$= \frac{2}{(1-\beta)^2} \int_a^1 \int_r^1 F(r)[1 - F(\tilde{r})]drd\tilde{r}$$

$$= \frac{2}{(1-\beta)^2} \int_a^1 (1 - p' + p'r)(\frac{1}{2}p' - p'r + \frac{1}{2}p'r^2)dr$$

$$= \frac{1-\beta}{3\beta + 1}12p'^2$$

We examine $\beta = 0.9$ (90%-CVaR) and when the range of $p'$ is $[0, p]$, where $\beta \leq 1 - p$. Taking the maximum $\sigma^2(\psi, F)$, which occurs precisely when $p' = p$, we obtain the variance, which allows us to construct the following confidence width that applies for any choice of shift-inducing $\lambda$ and threshold $\lambda'$:

$$c(n, \delta') = \Psi^{-1}(1 - \delta'/2)\frac{1}{1-\beta}\sqrt{\frac{(4 - 3p)p}{12n}}.$$

**Further experimental results**    Here, we include supporting plots to Fig. 3. Fig. 8 shows the proportion of runs with $\alpha - \Delta\alpha \leq R(\hat{\lambda}_T) \leq \alpha$, Fig. 9 shows sample trajectories for different values of $\tau$, analogous to the top two plots of Fig. 2, and Fig. 10 displays the risk trajectory for different selections of $\tau$.

### B.3    Risk Control in Demand Forecasting and Generation

In this synthetic experiment, we model a power plant that determines electricity generation at each timestep based on demand forecasts and a risk-control policy. The objective is to minimize overproduction while limiting the amount of unmet demand. Performativity arises through demand elasticity: surplus electricity can be stored in batteries and later used to offset shortages, linking one timestep's risk control decisions with future demand.

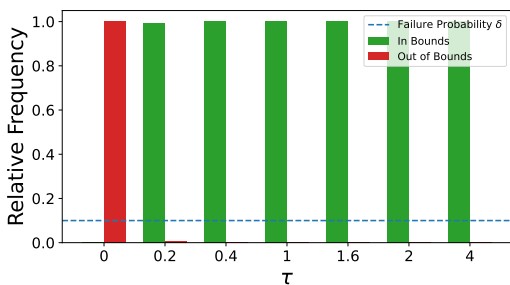

Figure 8: The relative frequency of runs that end with a $\hat{\lambda}_T$ with $\alpha - \Delta\alpha \leq R(\hat{\lambda}_T) \leq \alpha$ for quantile risk control in credit scoring.

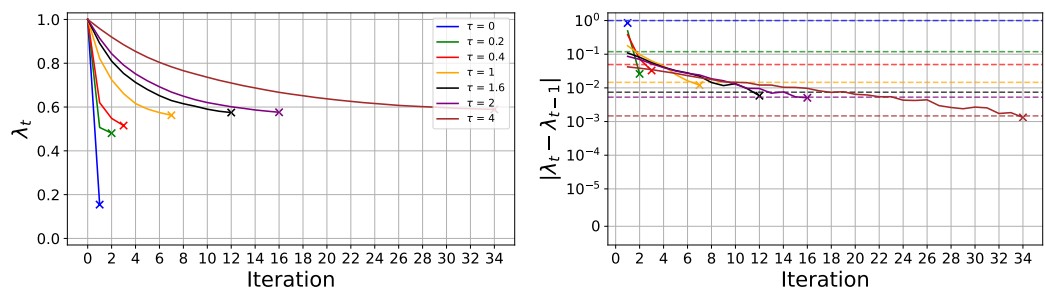

Figure 9: Analog of the bottom left plot in Fig. 2 for $\tau \in \{0, 0.2, 0.4, 1, 1.6, 2, 4\}$ in quantile risk control.

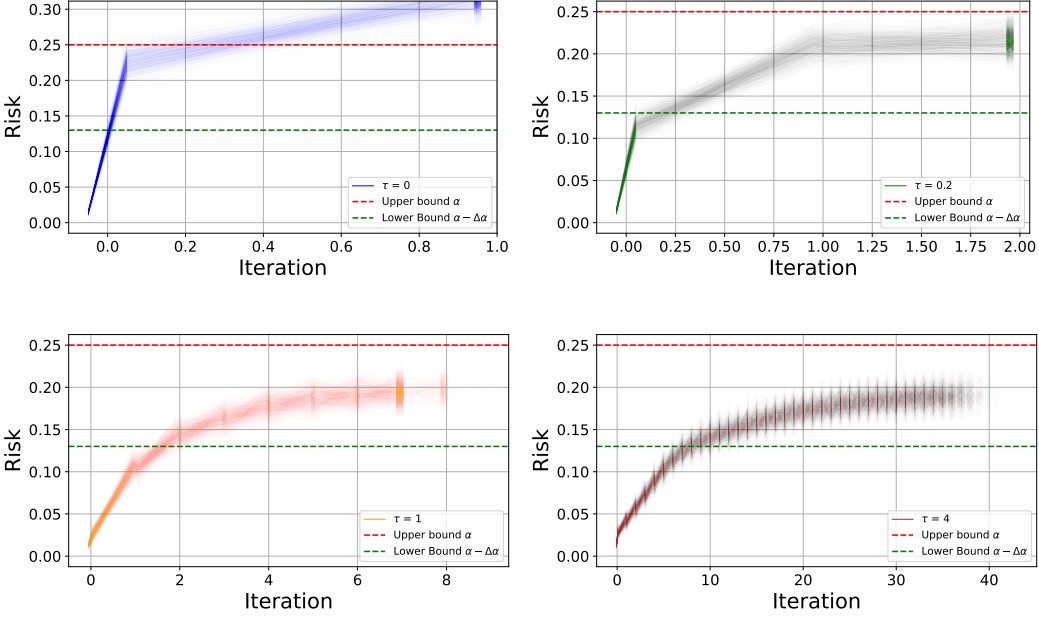

Figure 10: Analog of the bottom left plot in Fig. 2 for the $90\%$-CVaR risk measure and $\tau \in \{0, 0.2, 1, 4\}$.

**Data-generating process.** We simulate electricity demand at 10-minute resolution from 2005–2024 using a smooth, periodic model with additive Gaussian noise. Letting $s$ denote the timestamp, we define the normalized daily phase as

$$\phi_d(s) = \frac{s - \mathrm{daystart}(s)}{24\,\mathrm{h}},$$

where $\mathrm{daystart}(s)$ is 00:00 of the same calendar day. We simulate the demand process as follows:

$$y_s = \mu + A\sin\big(2\pi(\phi_d(s) - 0.5)\big) + \varepsilon_s, \qquad \varepsilon_s \sim \mathcal{N}(0, \sigma^2),$$

with $\mu = 30{,}000$, $A = 15{,}000$, $\sigma = 1{,}000$; the phase shift aligns peak demand at 6 PM. We then train a random forest regressor $f_\theta$ on data up to the end of 2007, using sine-cosine encodings of the hour of day and monthly phase as features $x_s$ with the target as $y_s$.

**Setting** At timestep $s$, the regressor forecasts demand $\hat{y}_s = f_\theta(x_s)$. The power plant then generates $U_s(\lambda) = (1 + \kappa\lambda)\hat{y}_s$ amount of electricity, where $\kappa$ is determined from a validation set (explained later) and $\lambda \in [0, 1]$. We make the assumption that performativity can be modeled as follows:

$$y_s(\lambda) = y_s - \sum_{r=1}^{R} \beta^r \left(U_{s-r}(\lambda) - y_{s-r}\right),$$

where $y_s$ is the demand of timestamp $s$ in the absence of performative effects, $R$ specifies the number of past timesteps influencing current demand, and $\beta$ controls the degree to which excess or insufficient generation in the past affects current demand. For the sake of illustration, we simplify the model by setting $R = 1$. The power plant's goal is to control the amount of unmet demand while accounting for this performativity: the continuous loss function is given by $\ell(x_s, y_s; \lambda) = (y_s(\lambda) - (1 + \kappa\lambda)\hat{y}_s)_+$.

Next, we use validation data from 2008 to determine appropriate settings for $\kappa$, $c(n, \delta')$, and $\tau$. Note that in these calculations, we assume knowledge of the distribution mapping $\mathcal{D}(\lambda)$. While this knowledge can be exploited to directly calculate a threshold $\hat{\lambda}$ that achieves our desiderata from the first batch of samples, we deliberately use PRC for the sake of illustration.

**Evaluating $\kappa$, $c(n, \delta')$, and $\tau$ from the Validation Set** Next, we use validation data from 2008 to determine appropriate settings for $\kappa$, $c(n, \delta')$, and $\tau$. Note that in these calculations, we assume knowledge of the distribution mapping $\mathcal{D}(\lambda)$. While this knowledge can be exploited to directly calculate a threshold $\hat{\lambda}$ that achieves our desiderata from the first batch of samples, we deliberately use PRC for the sake of illustration.

First, $\kappa$ is chosen large enough so that risk is controlled at $\lambda_{\mathrm{safe}} = 1$, i.e.,

$$U_s(\lambda_{\mathrm{safe}}) - y_s(\lambda_{\mathrm{safe}}) \geq 0, \qquad (1 + \kappa)\,\hat{y}_s - y_s + \beta\big(U_{s-1}(\lambda) - y_{s-1}\big) \geq 0.$$

We select $\kappa$ such that at least 95% of the validation data satisfy this inequality. Next, variance for the CLT confidence width $c(n, \delta')$ is defined via

$$\mathrm{Var}\big(\ell(x_s, y_s; \lambda)\big) = \mathrm{Var}\Big(\left[y_s(\lambda) - (1 + \kappa\lambda)\hat{y}_s\right]_+\Big),$$

and by searching over a grid of possible $\lambda$ values, we obtain the maximum sample variance to use in the bound at confidence level $\delta'$. Finally, the Lipschitz bound $\tau$ is derived from the loss

$$\ell(x_s, y_s; \lambda) = \left[y_s(\lambda) - (1 + \kappa\lambda)\hat{y}_s\right]_+,$$

which satisfies the local Lipschitz property

$$\left|\ell(x_s, y_s; \lambda_1) - \ell(x_s, y_s; \lambda_2)\right| \leq \left[\kappa\hat{y}_s + \beta\kappa\hat{y}_{s-1}\right]|\lambda_1 - \lambda_2|.$$

We therefore set $\tau$ as the 95th percentile of $\kappa\hat{y}_s + \beta\kappa\hat{y}_{s-1}$ across the validation set.

**PRC Results** Finally, we run PRC on the rest of the data, calibrating on one month (more precisely, 28 days) and testing on the next. We run one trajectory starting on each quarter to simulate randomness. We use $\alpha = 300$, $\Delta\alpha = 85$, $\delta = 0.1$, $N = 4032$ (the number of 10 minute intervals in 28 days), and $\beta = 0.2$. On the validation data, $\kappa = 0.059$, the variance for the CLT bound is 373155, and $\tau = 3204.43$.

Fig. 11 demonstrates our results. While the non-performative algorithm ($\tau = 0$) exceeds the risk threshold on the first iteration, the performative algorithm ($\tau = 3204.43$) carefully reduces $\lambda_t$ as to maintain risk control.

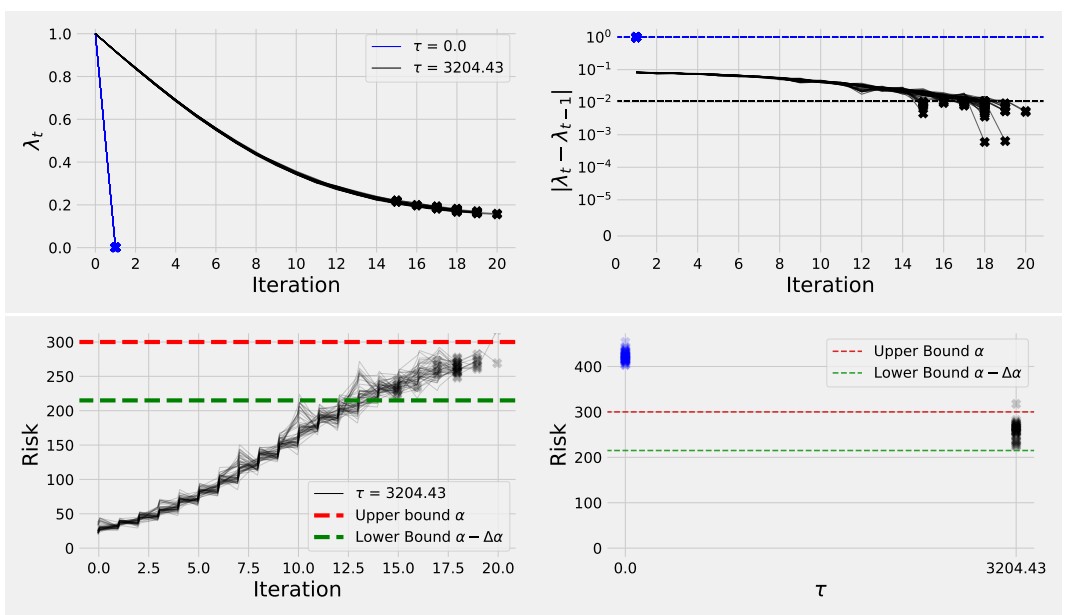

Figure 11: Unmet demand error control. We use the same plotting scheme as in Fig. 2.

