# OpenReview forum: "Performative Risk Control: Calibrating Models for Reliable Deployment under Performativity"
_NeurIPS.cc/2025/Conference — NeurIPS 2025 poster_

### Official Review · Reviewer_zDgW · 2025-06-28

**Clarity:** 3
**Significance:** 3
**Originality:** 2
**Rating:** 4
**Confidence:** 3

**Summary:**

The paper proposes a new framework to calibrate blackbox prediction models in environments where predictions influence future data distributions—a setting known as performativity. Building on recent work in risk control and conformal prediction, the authors extend these methods to account for performativity. They develop an iterative algorithm that selects a sequence of decision thresholds $\lambda_t$, guaranteeing both finite-sample statistical risk control and eventual tightness around a user-specified target risk level $\ Delta\alpha$, despite the distributional shifts induced by previous deployments. The approach is demonstrated through experiments in credit risk scoring, under both expected risk and quantile-based risk measures like CVaR.

**Questions:**

1. Monotonicity Assumption: The assumption that $\ell(z;λ)$ is non-increasing in $\lambda$ seems restrictive and significantly simplifies the problem. It effectively encodes a unidirectional notion of risk, implying that the decision-maker is only penalized for type II errors (e.g., false negatives), and that increasing conservativeness always reduces loss. This overlooks settings where there are different types of errors where non-monotonic losses arise, and make the search for an optimal threshold $\lambda$ much easier.

2. Motivation for loosening the threshold (Section 2.2): The rationale for choosing a less conservative (i.e., smaller) threshold $\lambda$ is stated in terms of "preserving utility" and "useful class information,"  (line 97) but these terms are not clearly defined. It would be helpful to clarify what utility means in this context -- what constitutes "useful class information" (which seems to be not captured by the loss $\ell(z; \lambda)$?) More broadly, a more detailed discussion of the trade-off between safety (risk control) and utility would strengthen the motivation behind relaxing the threshold.

3. Tumor segmentation example: I find this example difficult to interpret. My understanding is that $\lambda$ represents a deployment parameter that influences both (a) the prediction set size (e.g., the number of pixels labeled as tumor), and (b) the data distribution in the next round via performative feedback. However, it is unclear how the induced distribution $D(\lambda)$ should be interpreted in this context. Does a larger $\lambda$ simply produce more inclusive prediction sets, or does it also affect how future data (e.g., future tumor scans or labels) are generated? A more detailed explanation of how performativity manifests in the tumor segmentation setting would help clarify this.

**Ethical Concerns:**

["NO or VERY MINOR ethics concerns only"]

**Final Justification:**

The paper is the first work to propose a provable risk control method under performativity, which is a valuable contribution to both communities.  A clarification of the additional technical contributions specific to the performative setting would be valuable.

**Limitations:**

Yes

**Quality:**

3

**Strengths And Weaknesses:**

## Strengths:

The paper addresses a crucial gap in the risk control literature—how to ensure safe deployment in settings where predictions causally influence future data. This is particularly relevant in real-world socio-technical systems like credit scoring or policy-making. The extension to quantile-based risk measures like CVaR is also valuable for applications that care about tail risk rather than average-case loss.


## Weaknesses:

The major weakness of the paper is the monotonicity assumption which greatly simplify the complexity of the problem. The paper mentions a trade-off between “safety” and “utility” but doesn't clearly define what utility means or how to measure it in practice. The tumor segmentation example used to motivate the method is difficult to interpret in a performative setting. It’s unclear how prediction set size (controlled by $\lambda$) influences future data distribution. In addition, it's a bit unclear how the method is specifically designed to account for "performativity" -- the proposed method seems to work for any case where the distribution is smooth and doesn't change dramatically? See more details in the question section.

---

> ### Author Rebuttal · Authors · 2025-07-30
>
> Thank you for the detailed feedback. We address your questions below and hope that our answers can ease your concerns.
>
> > **W1/Q1**: On the monotonicity assumption. "The assumption that $\ell(z;\lambda)$ is non-increasing in $\lambda$ seems restrictive and significantly simplifies the problem. It effectively encodes a unidirectional notion of risk, implying that the decision-maker is only penalized for type II errors (e.g., false negatives), and that increasing conservativeness always reduces loss. This overlooks settings where there are different types of errors where non-monotonic losses arise, and make the search for an optimal threshold $\lambda$ much easier."
>
> We adopt the monotonicity assumption for two primary reasons. First, monotonicity is commonly used and adopted in the risk control literature (see [6] *Conformal risk control*). Second, and more importantly, for the performative socio-technical systems we considered, a monotonic loss has important semantic meanings and makes great sense.
>
> To begin with, we would like to clarify the details regarding the monotonic assumption we make. Our goal is to control the risk $R(\lambda):=R(\lambda,\lambda)$. *We do not assume this quantity to be monotonic.* We agree that assuming this quantity to be monotonic would greatly simplify the complexity of the problem. Instead, we assume $R(\lambda,\cdot)$ is monotonic due to $\ell(z,\cdot)$ being monotonic in the second term, as you highlighted. The difference is that the risk may increase or decrease when we change distributions. If we have $R(\lambda,\lambda')$ where the data distribution arises from $\lambda$ and we use $\lambda'$ to evaluate, then we have no monotonicity assumption when $\lambda$ changes and assume monotonicity when $\lambda'$ changes.
>
> For example, in credit scoring, we imagined an institution gradually deploying an algorithmic model. This model is tasked with identifying low-risk opportunities and providing automatic credit approval to applicants. The threshold $\lambda$ controls to what extent this model is used. We saw the objective as maximizing efficiency by leveraging the model as much as possible while maintaining safety guarantees on it as we try to find an optimal level to deploy it. *Because we viewed the model as inherently risky, we also viewed its increased usage as increased risk, leading us to frame our problem with the monotonicity assumption.*
>
> However, we do agree that studying the non-monotonic case has its use cases. In the credit scoring example, the threshold $\lambda$ may define the balance between automatic acceptances by the credit model and the number of applications subjected to further human review. One might want to design a holistic loss function on this entire pipeline, which includes the errors made by the credit model, the errors made by human reviewers, and the time cost of human review. This loss function would not be monotonic.
>
> How would one attempt this problem? To clarify, one key aspect in safe deployment is "safe at any time." This aspect is both important for applications and makes the problem worthwhile to study (otherwise, one can just use binary search). Similar to the monotonic case that we study in the paper, one would have to (1) start from a "safe" $\lambda$ and (2) iteratively adjust $\lambda$ to optimize some objective while maintaining safety guarantees. (1) Without monotonicity, defining $\lambda_\text{safe}$ is not obvious. We would need other assumptions on the loss $\ell(z,\cdot)$ besides monotonicity or resort to domain expertise to define $\lambda_\text{safe}$. (2) However, afterwards, one can use a similar method to PRC. Let $U(\lambda)$ be the utility function defined on the domain of $\lambda$ (in our paper, we took this to be monotonically decreasing in $\lambda$), and assume we have deployed $\hat\lambda_{t-1}$. We should choose $\hat\lambda_t$ in a way that maximizes the utility function $U(\lambda)$ while controlling risk by reasoning about $R(\hat\lambda_t)$ from the sample risk $\hat R(\hat\lambda_{t-1}, \cdot)$. **So we think the procedure under non-monotonicity is much the same.**
>
> In contrast to the non-monotonic setting, we do find a natural interpretation for $\lambda_\text{safe}$ in the monotonic case: an institution does not use their model, abstaining from predictions (e.g. the credit model makes no automatic acceptances). For adherence to the literature and interpretation reasons, we assumed monotonicity, although our algorithm should be amenable to adjustments to handle non-monotonicity.
>
> > **W2/Q2**: "The paper mentions a trade-off between “safety” and “utility” but doesn't clearly define what utility means or how to measure it in practice." "Motivation for loosening the threshold (Section 2.2): The rationale for choosing a less conservative (i.e., smaller) threshold $\lambda$ is stated in terms of "preserving utility" and "useful class information," (line 97) but these terms are not clearly defined. It would be helpful to clarify what utility means in this context -- what constitutes "useful class information" (which seems to be not captured by the loss $\ell(z;\lambda)$?) More broadly, a more detailed discussion of the trade-off between safety (risk control) and utility would strengthen the motivation behind relaxing the threshold."
>
> In credit scoring, and other use cases like automated resume screening and content filtering, **"utility" means efficiency**. When we have "useful class information," that information can help inform quicker decisions and increase efficiency. In credit scoring, the "useful class information" of classifying someone as creditworthy can allow a credit institution to automatically give them credit without the need for too much manual review. Hence, this "useful class information" helps preserves the utility of the model by helping us save time.
>
> What constitutes "useful class information?" In credit scoring, we might classify an applicant **{creditworthy}**, **{not creditworthy}**, or **{creditworthy, not creditworthy}** (e.g., we don't know). The first two constitute "useful class information." The last one doesn't.
>
> **On the trade-off between safety and utility**: we imagine $\lambda$ as a lever we pull in how much we utilize the model. As $\lambda$ decreases (the model is used more), safety is decreased and utility is increased because we cede control to the model and try to save time. That is why we seek the minimum threshold $\lambda$ so that we can maximize utility while maintaining safety guarantees (Sec 2.2). While we do not explicitly define how "utility" is *measured* since we believe that will vary on a case-by-case basis, we do motivate why we wish to minimize $\lambda$ in our algorithm (L97). We can update the manuscript for better clarity regarding this point.
>
> > **W3/Q3**: "The tumor segmentation example used to motivate the method is difficult to interpret in a performative setting. It’s unclear how prediction set size (controlled by $\lambda$) influences future data distribution."
>
> We apologize for the confusion. The point of the tumor segmentation example was to help slowly introduce performative risk control (PRC) by first introducing conformal risk control, which was the perspective of the authors when they began this work. For tumor segmentation, we agree that it is difficult to imagine how $\lambda$ may influence the distribution of future tumor scans and labels. We used this example to lead into the following paragraph (L27-32) in which we introduce risk control in the performative setting (PRC). There, we highlight PRC via our flagship example, credit scoring, where performativity does come into play. A bank that uses the calibration parameter $\lambda$ to inform its final algorithmic decision should expect to see some performative effects from using that parameter, e.g. due to individuals attempting to strategically update their features to their own advantage.
>
> >  **W4**: "In addition, it's a bit unclear how the method is specifically designed to account for "performativity" -- the proposed method seems to work for any case where the distribution is smooth and doesn't change dramatically?"
>
> Our analysis does indeed make the assumption that performative effects due to changes in $\lambda$ are "smooth," an assumption both key to our method and relied upon in the performative prediction literature (e.g. $\epsilon$-sensitivity as found in Def 3.1 of [7] *Performative Prediction*). Where performativity comes in is highlighted in L32-33. Standard risk control literature examine the risk $R(\lambda) := \mathbb{E}\_{(x, y) \sim \mathcal{D}} \left[\ \ell(y, \mathcal{T}\_\lambda(x))\ \right]$. We analyze the same expression with a small change $\mathcal{D} \rightarrow \mathcal{D}(\lambda)$. To control this quantity and account for performativity, our method specifically formulated Def. 3.1 and Assumption 3.2, which were instrumental to the proofs of our main results (Alg 1, Thm 3.3). In this manner, we specifically designed to account for performativity, and we did so by requiring that performative effects be smooth.

---

> > ### Comment · Reviewer_zDgW · 2025-08-05
> >
> > I thank the authors for their detailed response — my questions have been addressed, and I will update my score accordingly. I will also engage with the other reviewers during the remainder of the discussion period. My final comment (W4) aligns with those raised by another reviewer: the proposed method appears to be a relatively straightforward extension of the approach in [6]. A clarification of the additional technical contributions specific to the performative setting would therefore be valuable.

---

> > > ### Author Response · Authors · 2025-08-06
> > >
> > > Thank you for your feedback. Regarding technical contributions, we opted for standard methods to keep our analysis simple and realistic. However, we did find the need to resolve certain technical challenges pertaining to performativity:
> > >
> > > - One challenge was the iterative nature of performativity and the fact that we don't know what is the required number of iterations $T$. We resolved this challenge by creating a simple stopping rule parameterized by $\Delta\lambda$ and using a conservative confidence width $c(n,\delta/\tilde T)$ based on an upper bound $\tilde T$ on $T$. We showed how $\tilde T$ can be obtained (L2 of Alg 1) and constructed the algorithm so as to provide the user feedback on whether theoretical guarantees for risk control can be accomplished *before the first sample of the initial distribution* is even collected. This property is important because it strengthens the algorithm's practicality.
> > >
> > > - Another challenge was getting bounds like Bernstein's inequality and CLT to work for an unknown distribution, which we demonstrate in App A.2. We would expect these bounds to be quite wide because they do not assume much on future distributions, yet we demonstrate how we can achieve reasonable bounds, as demonstrated in the credit scoring experiment.
> > >
> > > - We also note that we did find the need to reframe the definition of sensitivity (Def 3.1) compared to [7] (Performative Prediction). There, sensitivity was framed on the feature-labels $z$, while we framed sensitivity on the loss to alleviate a Lipschitz condition on the loss function.
> > >
> > > As we pointed out in one of our responses to another reviewer, we see these key technical contributions as supporting the utility and practicality of our main contribution---highlighting the critical area of risk control under performativity.

---

### Official Review · Reviewer_5vJ9 · 2025-07-01

**Clarity:** 4
**Significance:** 3
**Originality:** 3
**Rating:** 4
**Confidence:** 3

**Summary:**

This paper introduces performative risk control, a framework that post processes model predictions to achieve provable risk control under performativity. The proposed procedure is an iterative calibration process, and is shown to be safe at anytime and tight when the performative distribution $D(\lambda)$ is Lipschitz continuous wrt to $\lambda$ and a valid confidence width $c(n,\delta)$ (which can be obtained via e.g., Bernstein's inequality or CLT) is provided. Experiments on the Kaggle credit scoring dataset demonstrate the effectiveness of the proposed framework in predicting credit default risk.

**Questions:**

1. In the experiments on the Kaggle credit scoring dataset, the authors assume the performativity---dependent on $\lambda$ ---affects the data distribution through impacting the prediction function $f(x(\lambda))$ in a specific way given at the bottom of page 8. I wonder if the theoretical guarantees hold only when the Kaggle dataset satisfies this assumption, and what would happen if the assumption is violated.

**Ethical Concerns:**

["NO or VERY MINOR ethics concerns only"]

**Final Justification:**

I would recommend to accept this paper.

**Limitations:**

1. The PRC method requires access to i.i.d. samples from $D(\widehat{\lambda}_t)$ for each $t \in [T]$. In practice, collecting such data can be expensive and time-consuming, as it involves deploying $T$ models sequentially and gathering $n$ data points from each model before proceeding to the next. The paper would benefit from a discussion of this limitation and potential strategies for mitigation.

2. As mentioned before, it would be beneficial for the authors to clarify the technical novelty of this work.

**Quality:**

3

**Strengths And Weaknesses:**

The paper is well-written. The assumptions and theorems are clearly stated and the consequences and limitation are discussed. This is the first work that proposes a provable risk control method when the data exhibit performativity.

On the other hand, the proposed method seems to be a straightforward generalization of that in \[1] for i.i.d. data to the setting where performativity exists, since for each $\widehat\lambda_t$, the authors assume access to $n$ i.i.d. samples from $D(\widehat\lambda_t)$. The technical contribution is less clear.

[1]. Bates, S., Angelopoulos, A., Lei, L., Malik, J., & Jordan, M. (2021). Distribution-free, risk-controlling prediction sets. Journal of the ACM (JACM), 68(6), 1-34.

---

> ### Author Rebuttal · Authors · 2025-07-30
>
> Thank you for providing thoughtful, helpful, and positive feedback. We address your questions below and hope that our answers can further ease your concerns.
>
> > **W1/L2** Technical novelty
>
> Since our paper is the first to study risk control in the performative setting to our knowledge, we opted for standard methods to keep our analysis simple and realistic. However, in the process, we did find the need to resolve certain technical challenges.
>
> - The main challenge lay in extending risk control to an iterative setting *in which we do not assume knowledge* of the required number of iterations $T$. We opted to resolve this challenge using simple techniques, such as creating a simple stopping rule parameterized by $\Delta\lambda$ and using a conservative confidence width $c(n, \delta/\tilde T)$ based on an upper bound $\tilde T$ on $T$.
> - A second challenge was getting bounds like Bernstein's inequality and CLT to work *for an unknown distribution*, which we demonstrate in App A.2. We would expect these bounds to be quite wide because they do not assume much on future distributions, yet we demonstrate how we can achieve reasonable bounds, as demonstrated in the credit scoring experiment.
> - We also note that we did find the need to reframe the definition of sensitivity (Def 3.1) compared to [7] (*Performative Prediction*). There, sensitivity was framed on the feature-labels $z$, while we framed sensitivity on the loss to alleviate a Lipschitz condition on the loss function.
>
> In summary, we agree that most of our methods are standard. However, we see that our main contribution is highlighting a critical area of study, in which the safe and gradual deployment of societal algorithms must account for performative effects.
>
> > **Q1**: "In the experiments on the Kaggle credit scoring dataset, the authors assume the performativity---dependent on $\lambda$---affects the data distribution through impacting the prediction function $f(x(\lambda))$ in a specific way given at the bottom of page 8. I wonder if the theoretical guarantees hold only when the Kaggle dataset satisfies this assumption, and what would happen if the assumption is violated."
>
> Our distribution mapping $f(x(\lambda))$ is meant to show one example of an adversarial distribution shift in which individuals will shift across the threshold $1-\lambda$ if they are sufficiently close to it. In Appendix B.2, we show that this simulated shift is $(\gamma,1,\ell(\cdot,\lambda))$-sensitive for all $\lambda$, where we resolve $\gamma\leq1.38$. Hence, $f(x(\lambda))$ is just one instantiation of a class of distribution mappings that satisfy Assumption 3.2, which is a key assumption for achieving theoretical guarantees (Thm 3.3). The value $\gamma$, when compared to the safety parameter $\tau$, determines whether theoretical guarantees can be achieved. If $\tau\geq \gamma$ is *violated*, we may or may not achieve tight risk control, as shown in the bottom-right plot of Fig. 2. For example, $\tau\in\{0.2,0.5,0.8\}$ work, but $\tau\in\{0.001,0.1,0.2\}$ don't.
>
> > **L1**: "The PRC method requires access to i.i.d. samples from $D(\hat\lambda_t)$ for each $t\in[T]$. In practice, collecting such data can be expensive and time-consuming, as it involves deploying $T$ models sequentially and gathering $n$ data points from each model before proceeding to the next. The paper would benefit from a discussion of this limitation and potential strategies for mitigation."
>
> We agree with you that this is a limitation. However, this limitation mainly stems from inheriting the sampling requirements from conformal prediction and the iterative nature of performative algorithms.
>
> We note that for the applications we had in mind---credit scoring, automatic resume screening, content moderation---we did not expect $T$ to be very large. Indeed, in our credit scoring experiment, we had $T \approx 10$. With $N\approx 2000$, we would require $20,000$ samples (and not all at once but over some period of time).
>
> Further, knowledge of the distribution mapping $\mathcal{D}(\cdot)$ would help to mitigate the sample requirements of PRC. Indeed, if we could exactly parameterize $\mathcal{D}(\cdot)$, we could directly calculate $\hat R_n(\lambda,\lambda):=\frac{1}{n}\sum_{i=1}^n \ell(z_i(\lambda),\lambda)$, the finite-sample approximation of $R(\lambda)$, from the initial samples $\{z_i(\lambda_\text{safe})\}_{i=1}^n$ and calibrate directly based on this initial cohort. We would not need to iterate and could keep $T=1$. However, our analysis focused on unknown distribution shifts, as is often the case in practice.

---

> > ### Comment · Reviewer_5vJ9 · 2025-08-04
> >
> > I appreciate the authors for their detailed response. I will maintain my positive score.

---

### Official Review · Reviewer_hboG · 2025-07-01

**Clarity:** 3
**Significance:** 3
**Originality:** 3
**Rating:** 4
**Confidence:** 4

**Summary:**

The paper introduces a procedure for risk control in performative settings. The basic idea is to combine the model and assumptions from performative prediction with conformal risk control to obtain a method that controls risk while allowing for performative distribution shifts. The procedure is shown to be valid and it is evaluated in a semi-synthetic credit scoring experiment.

**Questions:**

Suppose $D(\lambda)$ comes from some known parametric model, or we have some other knowledge of the distribution shifts. How would your procedure change in that case? Can you give a more powerful version of the method?

How can you ensure $\Delta\alpha$ tightness for any value? I'm thinking of a setting where the risk function is discontinuous and has sharp jumps. In that case you shouldn't be able to get arbitrarily close to $\alpha$?

Can you comment more on the role of $\tilde T$? What would happen if we didn't have the condition $\Delta\lambda \geq (\lambda_{\mathrm{safe}} - \lambda_{\min}) / \tilde T$?

**Ethical Concerns:**

["NO or VERY MINOR ethics concerns only"]

**Final Justification:**

Overall I am supportive of acceptance. I believe more high-level discussions would be help, and the method is not exactly simple, but this is a good problem and a good first stab at a solution.

**Limitations:**

Yes

**Quality:**

3

**Strengths And Weaknesses:**

Studying risk control under performativity is interesting and novel to the best of my knowledge. The paper gives well-motivated example setting, such as the lending problem. Given that this is a new problem setting, I wish there was more high-level discussion on what are the desiderata in this problem (e.g. we probably care about what the induced distribution looks like). In general, more conceptual discussions would help the exposition. The procedure (Alg. 1) makes sense, though it would be nice to give some instantiations where some steps simplify (e.g. step 2).

Line 102: This should be with high probability.
Line 118: You say: "Notice that the calibration dataset ... in Def. 1.1 is the union of..." This is not stated in Def. 1.1.

---

> ### Author Rebuttal · Authors · 2025-07-30
>
> Thank you for the thoughtful and positive feedback. We address your questions below and hope that our answers can ease your concerns.
>
> > **W1**: Desiderata in this problem
>
> Thanks for pointing out the novelty of our setting. For desiderata,  we refer you to the high-level desiderata and significance of performative risk control (PRC) in Sec. 2.2. To be more concrete, in the example of credit scoring, we imagine an algorithmic model gradually being deployed to customers. This model is tasked with identifying low-risk opportunities and automatically accepting customers for credit loans. Initially, $\lambda$ is set so that only the safest customers get automatically approved. We continue to monitor the model's risk as we continue lowering $\lambda$, which expands automatic credit acceptance to more customers.
>
> Our desiderata are to "achieve user-specified conservativeness" (L97) and "safe at any time" (L101). As applied to credit scoring, the first statement means we utilize the model as much as we can, which is useful for both reducing manual work and expanding credit access. The second statement means that we hope to control the risk of unpaid loans, which is bad for both parties. Our work seeks to answer how we can maximally utilize the model in a performative setting while ensuring safety.
>
> **On "we probably care about what the induced distribution looks like"**: Ideally, we would like to make Alg. 1 applicable to all sorts of induced distributions. Sometimes, the induced distributions may actually lower risk, which would be the case if the threshold $\lambda$ is polarizing, making the classification task easier. In most cases, the induced distributions may be antagonistic, which we explored in the credit scoring experiment where applicants attempt to strategically manipulate their features. This setting is reasonable for other applications besides credit scoring, e.g., automatic resume screening and content moderation. To keep our analysis tractable, we impose a smoothness constraint on the distribution shift as is common in the literature. We hope that this explanation adequately addresses your concern.
>
> > **W2**: "more conceptual discussions would help the exposition"
>
> Thanks for the suggestion. In our future revision, we plan to illustrate our desiderata more with credit scoring as an example and highlight PRC's applicability in other settings like automatic resume screening and content moderation, as in our response to **W1**.
>
> > **W3**: "The procedure (Alg 1) makes sense, though it would be nice to give some instantiations where some steps simplify (e.g. step 2)"
>
> Alg. 1 was designed with two items in mind: simplicity and utility to the user. We do agree that step 2 is complex and will attempt to clarify why this level of complexity is needed. The answer ties into your questions **Q1** and **Q3**. The most relevant bits are in the next two paragraphs.
>
> As mentioned in our response to **Q1**, having more information on $D(\lambda)$, e.g. knowing its parameterization, dramatically simplifies the problem. In that case, we would not need Alg. 1 to be an iterative algorithm, can avoid the complexity of step 2, and can obtain $\hat\lambda$ on the first go.
>
> As pointed out in **Q3**, the role of $\tilde T$ is quite important. PRC requires knowledge of a confidence width on the first iteration (and all others), which in turn requires knowledge of the number of iterations $T$ (originally unknown). This is because to prove high probability guarantees for the entire trajectory, we use high probability guarantees on each individual step and *somehow have to compound those error rates.* Since the error rate increases with $T$, we faced a design choice (see **Q3** for more details). Either we guarantee an error rate that increases with $T$, or we increase some complexity in Alg 1. We saw the first option as undesirable to a user; it would be better to know the maximum error rate $\delta$ of the entire procedure before starting, so we went with the second option. Indeed, doing so also loosens the bounds; if we had knowledge of $T$, we could use a confidence width $c(n,\delta/T)$ rather than $c(n,\delta/\tilde T)$. However, we saw the trade-off as worthwhile, since for most bounds we considered, the difference would have been a small factor similar to $\sqrt{ \frac{\ln(2\tilde T/\delta)}{\ln(2T/\delta)} }$ (computed from Hoeffding's). Hence, the complexity of step 2 stems from a design choice aimed at maximizing PRC's utility to the user: we believe the algorithm should inform the user whether theoretical guarantees are possible *before* data collection and believe this benefit warrants the complexity of the joint-solve in Alg. 1 L2.
>
> > **W4**: L102 should be with high probability
>
> Yes, we will revise this line to be clearer. Thanks for the reminder.
>
> > **W5**: L118 "Notice that the calibration dataset ... in Def. 1.1 is the union of..."
>
> Yes, $\mathcal{I}\_\text{cal}=\cup_t \mathcal{I}\_\text{cal}^t$. We decided to leave out this detail in Def. 1.1 for exposition purposes. The goal of PRC is to find a desirable $\hat\lambda$, and Def. 1.1 is meant to illustrate our work's alignment with the broader risk control literature. For this reason, we chose to keep the exposition simple by just writing $\mathcal{I}\_\text{cal}$.
>
> > **Q1**: Can we give a more powerful version of our method supposing that $D(\lambda)$ comes from some known parametric model?
>
> Yes, if $D(\lambda)$ could be modeled completely, we would not need to make our algorithm iterative. We would then be able to calculate $\hat R_n(\lambda,\lambda):=\frac{1}{n}\sum_{i=1}^n \ell(z_i(\lambda),\lambda)$, the finite-sample approximation of $R(\lambda)$, directly off of the initial samples $z_i(\lambda_{\text{safe}})$ for $i = 1,\dots,n$ and calibrate directly based on the initial cohort. We point out that our distribution mapping in the credit scoring experiment is deterministic and amenable to this version of the method; however, we aimed to keep our analysis focusing on unknown distribution shifts. To keep the analysis tractable, we follow the literature in imposing a Lipschitz constraint on our distribution shifts. Here, the iterative method comes into play because we do not assume that we know $z\_i(\lambda)$ directly from $z\_i(\lambda\_\text{safe})$.
>
> > **Q2**: How can you ensure $\Delta\alpha$ tightness for any value? ... In that case you shouldn't be able to get arbitrarily close to $\alpha$?
>
> That's correct. This is where the assumption of loss **continuity** is critical. In Thm 3.3, we state that the loss function $\ell(z,\lambda)$ must be continuous in $\lambda$ for all $z$. This assumption allows us to get our risk arbitrarily close to $\alpha$. For the setting where the risk function is discontinuous and has sharp jumps, one can refer to Sec 2.2 of [6] (*Conformal Risk Control*).
>
> > **Q3**: The role of $\tilde T$ and what happens if we didn't have the condition $\Delta\lambda \geq (\lambda_\text{safe} - \lambda_\text{min})/\tilde T$
>
> We defined $\tilde T$, an upper bound on the number of iterations $T$, so that we could calculate the confidence width $c(n,\delta/\tilde T)$ at the beginning. The advantage of this method is that the PRC algorithm can give the user feedback on whether their specified level of tightness $\Delta\alpha$ can be theoretically guaranteed *before any samples are collected,* as detailed in our response to **W3**.
>
> If we didn't have the condition $\Delta\lambda \geq (\lambda_\text{safe} - \lambda_\text{min})/\tilde T$, or $\tilde T \geq (\lambda_\text{safe} - \lambda_\text{min})/\Delta\lambda$, we may not be able to achieve user-specified conservativeness, especially if the optimal $\hat\lambda$ is close to $\lambda_\text{min}$ and each iteration decreases $\lambda$ by barely more than $\Delta\lambda$. In this case, $\lambda$ may not "make it in time" to $\hat\lambda$. It is still possible that PRC terminates on a good $\hat\lambda$, but with no theoretical guarantees.

---

> > ### Comment · Reviewer_hboG · 2025-08-05
> >
> > Thank you for the detailed and thoughtful response. I don't fully understand your response to the question about having a parametric model for $D(\lambda)$. Can you clarify why you would be able to calculate $R_n(\lambda, \lambda)$ for *any* $\lambda$ given only initial samples from $\lambda_{\text{safe}}$? I believe this claim makes some implicit assumptions. I am not referring to the case where the map is fully known, just the case where it can be fit parametrically.

---

> > > ### Author Response · Authors · 2025-08-06
> > >
> > > Thank you for the feedback. Yes, apologies for the confusion: we interpreted the initial question as a distribution map whose parameterization *and* parameters are fully known.
> > >
> > > For a parametric mapping with unknown parameters, one might try to improve the convergence rate by using the samples gathered during deployments to not only select the next $\hat\lambda_t$ but also to make estimations on the mapping parameters. In [8] *Outside the Echo Chamber*, the authors construct a two-stage process whose first stage includes randomly sampling classifier parameters $\theta$ and observing its effects on the distribution mapping $D(\theta)$. The second stage then minimizes the finite-sample approximation of the performative risk. The parallel in our case would be randomly sampling $\lambda$ so as to learn the mapping $D(\lambda)$. Once we know the parameterization of our mapping, we could calculate $\hat R_n(\lambda,\lambda)$ and calibrate from there.
> > >
> > > However, random sampling is not practical since it violates our desiderata "safe at any time." We would need to make estimations on the mapping parameters while choosing $\hat\lambda_t$ according to Alg 1. However, if we were to follow Alg 1, we would *not* be collecting samples from randomly selected $\lambda$'s and may run into rank deficiency issues in parameter estimation. In that case, there is not much more we can do. If somehow rank deficiency is not an issue and we could perform mapping parameter estimation, we could start to rely on our knowledge of $D(\lambda)$ more in the later iterations to estimate $\hat R_n(\lambda,\lambda)$ and update $\lambda$ more aggressively. We hope this explanation addresses your question.

---

> > > > ### Comment · Reviewer_hboG · 2025-08-07
> > > >
> > > > Thank you for the explanation. This was helpful.

---

### Official Review · Reviewer_gYio · 2025-07-06

**Clarity:** 2
**Significance:** 2
**Originality:** 2
**Rating:** 4
**Confidence:** 3

**Summary:**

The paper studies risk control in the performative setting, e.g. where individuals/organizations/datapoints will strategically attempt to manipulate their features so as to influence risk control variables such as the size of the output set that the model assigns to them. Specifically, the authors adapt a standard iterative approach to performative prediction for risk control, which they refer to as Performative Risk Control. The algorithm follows a common iterative approach to performative prediction: in this case, starting with a conservative threshold, collecting new data from the distribution induced by the previous threshold, building an upper confidence bound on the new threshold (accounting for potential distribution shift, and assuming smoothness as is standard), and repeating. The proof of the main theorem arguing the convergence of the method (which is entirely deferred to the Appendix without even a sketch) implements the usual performative prediction proof approach by adapting UCB to select updated thresholds; the convergence rates and assumptions match exactly what one would expect from standard performative prediction results.

**Questions:**

Questions:
- Addressing the weaknesses would be helpful in making me feel comfortable with a higher score.
- Normally, performative prediction is motivated by individuals/organizations that want to bias their own prediction in a certain direction. However, risk control is normally applied to higher order statistics, most commonly uncertainty set size. The degree to which individuals/organizations are incentivized to manipulate features to influence such stats is less clear; I don't doubt that such a motivation exists, but what are some concrete examples?
- What does poly mean on L25?

**Ethical Concerns:**

["NO or VERY MINOR ethics concerns only"]

**Final Justification:**

I maintain a bias towards acceptance.

**Limitations:**

Yes

**Quality:**

3

**Strengths And Weaknesses:**

Strengths:
- The authors argue for an interesting problem and while the paper could better motivate performative risk control, it's not hard to see why this may be of interest (e.g. why a suspect may hope that a judge forms a wider uncertainty set for guilt).
- The actual theoretical result (which is the main contribution) largely follows a standard recipe for adapting a learning problem to the performative setting but its nice to see the details worked through; the form of the algorithm, assumption, and rates appear reasonable.

Weaknesses:
- The current discussion for motivating performative risk control leaves something to be desired; see questions. This is important because, while the theorem proof isn't very involved and mainly just follows a standard recipe, the paper could still have a meaningful contribution in arguing for the study of performative prediction + conformal provided the problem is better motivated.
- Writing can be generally improved, especially in technical parts. For example, the statement of Theorem 3.3 is a hard to read: delta alpha isn't defined, you need to scan up to L163 to read it instead; same thing with Algorithm 1 (esp. Line 2). I don't believe that the lack of clarity is unavoidable (the algorithm is not too complicated, though the authors may find it cleaner to reparameterize $\Delta \alpha$ and $c(\cdot)$ ?), just that the writing just needs more polishing.

Comments:
- It may be worth editing Figure 2 so that tau = 0.001 is labeled as the non-performative algorithm (or more precisely to include one where you exactly ablate the performative aspect with tau = 0). Given the way the experiments are setup (distribution shift is synthetically added exactly so that these types of performative prediction algorithms work well), the result of the experiment seems somewhat expected, but it's still nice to have experimental validation that the paper's implementation for risk control does work.
- While the existing related work discussion covers performative prediction fairly thoroughly, the authors may consider also including discussion on the connections to Stackelberg literature (though this is a point of broader friction between the two communities rather than a shortcoming of this paper).

---

> ### Author Rebuttal · Authors · 2025-07-30
>
> Thank you for the thoughtful and detailed feedback. We address your questions below and hope that our answers could ease your concerns.
>
> > **W1/Q2**: Motivation and concrete examples
>
> Performative risk control (PRC) is motivated by the phenomenon of **performativity** in decision-making. Since the goal of (conformal) risk control is to calibrate any black-box model by choosing low-dim calibration parameters like thresholds to ensure reliable decisions, the phenomenon of performativity will naturally happen---the target distribution will change according to the chosen parameters. This is particularly prevalent in applying risk control in many socio-technical systems. This finding is the motivation behind our paper. We provide credit scoring as an example in L27-32, where a bank's loan policies set by $\lambda$ change the purchasing patterns of the population, and hence the features used in loan prediction itself.
>
> To be more comprehensive, we here provide additional concrete motivating examples.
>
> **Resume screening example.** A company may set $\lambda$ to determine the percentage of applicants to screen while controlling the risk of screening out solid applicants. Formally, we can interpret the calibrated prediction $\mathcal{T}\_\lambda(x)$ in one of two ways. One, $\mathcal{T}\_\lambda(x)$ is an uncertainty set that can take one of the following values:
> **{accept}**, **{reject}**, and **{accept, reject}**. Since the purpose of resume screening isn't to automatically accept applicants, **{accept}** doesn't make sense as an option. Hence, $\mathcal{T}\_\lambda(x)$ is either **{reject}** or **{accept, reject}**: either the applicant is automatically rejected or the screener is uncertain which class the applicant belongs to, and the applicant passes onto the next round. Seen in this way, we can also view $\mathcal{T}\_\lambda(x)$ as a point prediction, in which the applicant is either automatically rejected or passes onto the next round. In both interpretations, the decision made is the same, showing how the application of risk control to higher-order statistics affects real decisions and incentivizes individuals to manipulate their features to their advantage. Concretely, the performativity aspect in this example comes from screening rigor impacting how applicants apply, e.g., to what extent an applicant "AI-proofs" their resume or optimizes keywords.
>
> **Content modification example.** A social media platform may set $\lambda$ to determine how much to censor. This, in turn, impacts what users post. For example, in response to more rigorous censoring, they may try to do more "clean up" on their post to ensure it is not censored. Hence, deployment of $\lambda$ impacts the distribution of posts. At your suggestion, we can include these examples in our exposition.
>
> In summary, we find performative risk control to be highly relevant to socio-technical systems, and our work provides an algorithm and theoretical framework for safely deploying models within these dynamic social environments.
>
> > **W1**: Technical contributions
>
> We appreciate the reviewer pointing out that one important contribution of this work is **initiating** the study of performativity in the setting of risk control. Meanwhile, we want to point out that, on the technical side, our theorem proof is standard to keep our analysis as simple and realistic as possible.
>
> However, our setting has particular technical challenges. Specifically, we formally stated new desiderata and the significance of our framework in Sec. 2.2, which required us to carefully design an algorithm to satisfy them. Meanwhile, in extending risk control to the performative case, we found the need to reframe the definition of sensitivity (Def. 3.1) and deal with a confidence bound that grows with the number of iterations. In [7] (*Performative Prediction*), sensitivity was framed on the feature-labels $z$, while we framed sensitivity on the loss to alleviate a Lipschitz condition on the loss function. Further, our extension of risk control to an iterative setting---without presuming knowledge of $T$, the number of iterations---required resolving several technical challenges, which included the definition of $\tilde T$ and various bound calculations in Appendix A.2.
>
> > **W2**: Writing clarity
>
> **On Thm 3.3**: We agree and we will plan to rearrange the statement of this theorem in a more readable format.
>
> **On $\Delta\alpha$**: We point out that we had $\Delta\alpha$ defined as a user-controlled tightness parameter in the technical setup sections (Sec 2.2 L97-99, Sec 3 Intro L107, L114). Let us know if you think a reader might prefer a more explicitly defined $\Delta\alpha$ in the theorem itself.
>
> **On Alg 1 L2**: Yes, we agree that Line 2 is quite complex. In a future revision, we will try to make it less verbose, perhaps splitting it into two steps. However, we do believe that the joint-solve of $(\tilde T, \Delta\lambda)$ is crucial to our algorithm, and this complexity stems from a design choice in which we sacrificed simplicity for user utility.
>
> Specifically, our goal is to provide a high probability guarantee for the entire trajectory at the beginning *before any samples are collected.* To do so, we require knowledge of a confidence width on the first iteration (and all others), which in turn requires knowledge of the number of iterations $T$ (unknown). This is because to prove high probability guarantees for the entire trajectory, we use high probability guarantees on each individual step and *somehow have to compound those error rates.* Since the error rate increases with $T$, we faced a design choice. Either we guarantee an error rate that increases with $T$, or we increase some complexity in Alg 1 (e.g. L2). We saw the first option as undesirable to a user; it would be better to know the maximum error rate $\delta$ of the entire procedure before starting, so we took the second option. Indeed, doing so also loosens the bounds; if we had knowledge of $T$, we could use the confidence width $c(n,\delta/T)$ rather than $c(n,\delta/\tilde T)$. However, we saw the trade-off as worthwhile, since for most bounds we considered, the difference would have been a small factor similar to $\sqrt{ \frac{\ln(2\tilde T/\delta)}{\ln(2T/\delta)} }$ (computed from Hoeffding's). Hence, the complexity of step 2 stems from a design choice aimed at maximizing PRC's utility to the user: we believe the algorithm should inform the user whether theoretical guarantees are possible *before* data collection and believe this benefit warrants the complexity of the joint-solve in Alg. 1 L2.
>
> **On reparameterizing $\Delta\alpha$ and $c(\cdot)$**: $\Delta\alpha$ is user-specified. We imagine a user typing in what constant they want for $\Delta\alpha$, similar to how $\alpha$ is user-defined. $c(\cdot)$ is just a generalized shorthand referring to the confidence width, which depends on the number of samples $n$, the error rate $\delta'=\delta / \tilde T$, and the type of concentration bound used.
>
> > **C1**: editing Figure 2 so that tau = 0.001 is labeled as the non-performative algorithm
>
> Agree with both points. In the revision, we will make clear that tau = 0.001 corresponds to the algorithm that does not account for performativity.
>
> > **C2**: connections to Stackelberg literature
>
> Thanks for the suggestion. This work, especially the credit scoring experiment (which follows [7] *Performative Prediction*), has strong ties with the strategic classification and more broadly the Stackelberg literature. We can illustrate this connection more in our related works section.
>
> > **Q3**: "poly" on L25
>
> Sorry about the typo. We meant "polyp segmentation" as described in our reference [5]. We used this example to illustrate and motivate (conformal) risk control. Thank you for pointing this typo out.

---

> > ### Comment · Reviewer_gYio · 2025-08-06
> >
> > Thanks for the responses. The authors have addressed my questions (which were mostly not major concerns anyways) so I continue to lean towards acceptance.

---

### Decision · Program_Chairs · 2025-09-17

**Decision:**

Accept (poster)

**Comment:**

This paper introduces Performative Risk Control, a framework that addresses risk control in performative settings, defined in the paper as a situation prediction-supported decisions may influence the outcome they aim to predict. The authors extend existing risk control and conformal prediction methods to account for these performative effects. The method is validated through experiments, including a credit risk scoring application.

Reviewers found the motivation for the paper - risk control in performative settings to be interesting. The paper was well-written with clear assumptions and theorems, though some reviewers note that the technical contribution may be a relatively straightforward adaptation of existing methods to the performative setting. Reviewers also pointed out specific areas where the motivation and assumptions, such as the monotonicity assumption, could be better explained and justified. Nevertheless, the reviewers found the approach promising and the technical details convincing. The authors are encouraged to revise the paper in light of reviewer feedback.